# Cryptic transmission of SARS-CoV-2 and the first COVID-19 wave

Jessica T. Davis[1,11], Matteo Chinazzi[1,11], Nicola Perra[1,2,11], Kunpeng Mu[1], Ana Pastore y Piontti[1], Marco Ajelli[3], Natalie E. Dean[4], Corrado Gioannini[5], Maria Litvinova[3], Stefano Merler[6], Luca Rossi[5], Kaiyuan Sun[7], Xinyue Xiong[1], Ira M. Longini Jr[8], M. Elizabeth Halloran[9,10], Cécile Viboud[7] & Alessandro Vespignani[1✉]

Considerable uncertainty surrounds the timeline of introductions and onsets of local transmission of severe acute respiratory syndrome coronavirus 2 (SARS-CoV-2) globally[1–7]. Although a limited number of SARS-CoV-2 introductions were reported in January and February 2020 (refs.[8,9]), the narrowness of the initial testing criteria, combined with a slow growth in testing capacity and porous travel screening[10], left many countries vulnerable to unmitigated, cryptic transmission. Here we use a global metapopulation epidemic model to provide a mechanistic understanding of the early dispersal of infections and the temporal windows of the introduction of SARS-CoV-2 and onset of local transmission in Europe and the USA. We find that community transmission of SARS-CoV-2 was likely to have been present in several areas of Europe and the USA by January 2020, and estimate that by early March, only 1 to 4 in 100 SARS-CoV-2 infections were detected by surveillance systems. The modelling results highlight international travel as the key driver of the introduction of SARS-CoV-2, with possible introductions and transmission events as early as December 2019 to January 2020. We find a heterogeneous geographic distribution of cumulative infection attack rates by 4 July 2020, ranging from 0.78% to 15.2% across US states and 0.19% to 13.2% in European countries. Our approach complements phylogenetic analyses and other surveillance approaches and provides insights that can be used to design innovative, model-driven surveillance systems that guide enhanced testing and response strategies.

A few weeks after the initial announcement of a cluster of atypical pneumonia cases in Wuhan, China, the first confirmed cases of coronavirus disease 2019 (COVID-19) in the USA and Europe were detected (on 21 January 2020 in WA, USA[1] and on 24 January 2020 in France[2]). Although many more states and countries began to report initial introductions in the following weeks, only a few cases were detected daily during this time period (Fig. 1a), and most countries adopted a testing policy that targeted symptomatic individuals with a travel history linked to China. Several reports suggest that the introduction of SARS-CoV-2 occurred earlier than initially recognized[3–8], raising questions about the effectiveness of the initial testing policies and travel-related restrictions, as well as the extent to which the SARS-CoV-2 virus spread through cryptic transmission in January and February 2020. To address these questions, we use the global epidemic and mobility (GLEAM) model, a data-driven, stochastic, spatial, age-structured metapopulation epidemic model[11,12], to study the global dynamic underlying the evolution of the COVID-19 pandemic in Europe and the USA. Our model maps the plausible pathways of the pandemic using information available at the early stages of

the outbreak and provides a global picture of the cryptic phase as well as the ensuing first wave of the COVID-19 pandemic.

We consider data concerning the continental USA and 30 European countries (the full list is reported in Extended Data Table 1). The model integrates real-time human mobility and population data with a mechanistic epidemic model at a global scale, incorporating changes in contact patterns and mobility according to the non-pharmaceutical interventions (NPIs) implemented in each region. It is calibrated on international case introductions out of mainland China at the early stage of the pandemic using an approximate Bayesian computation (ABC) methodology[13]. The model returns an ensemble of stochastic realizations of the global epidemic spread including international and domestic infection importations, incidence of infections and deaths at a daily resolution (see Methods). In the following text, we provide a detailed discussion of the analyses and results concerning European countries and the US states; however, to further test and validate our approach, in the Supplementary Information, we report the modelling results for 24 additional countries that are globally representative,

[1]Laboratory for the Modeling of Biological and Socio-technical Systems, Northeastern University, Boston, MA, USA. [2]Networks and Urban Systems Centre, University of Greenwich, London, UK. [3]Department of Epidemiology and Biostatistics, Indiana University School of Public Health, Bloomington, IN, USA. [4]Department of Biostatistics and Bioinformatics, Emory University, Atlanta, GA, USA. [5]ISI Foundation, Turin, Italy. [6]Bruno Kessler Foundation, Trento, Italy. [7]Division of International Epidemiology and Population Studies, Fogarty International Center, National Institutes of Health, Bethesda, MD, USA. [8]Department of Biostatistics, College of Public Health and Health Professions, University of Florida, Gainesville, FL, USA. [9]Vaccine and Infectious Disease Division, Fred Hutchinson Cancer Research Center, Seattle, WA, USA. [10]Department of Biostatistics, University of Washington, Seattle, WA, USA. [11]These authors contributed equally: Jessica T. Davis, Matteo Chinazzi, Nicola Perra. ✉e-mail: a.vespignani@northeastern.edu

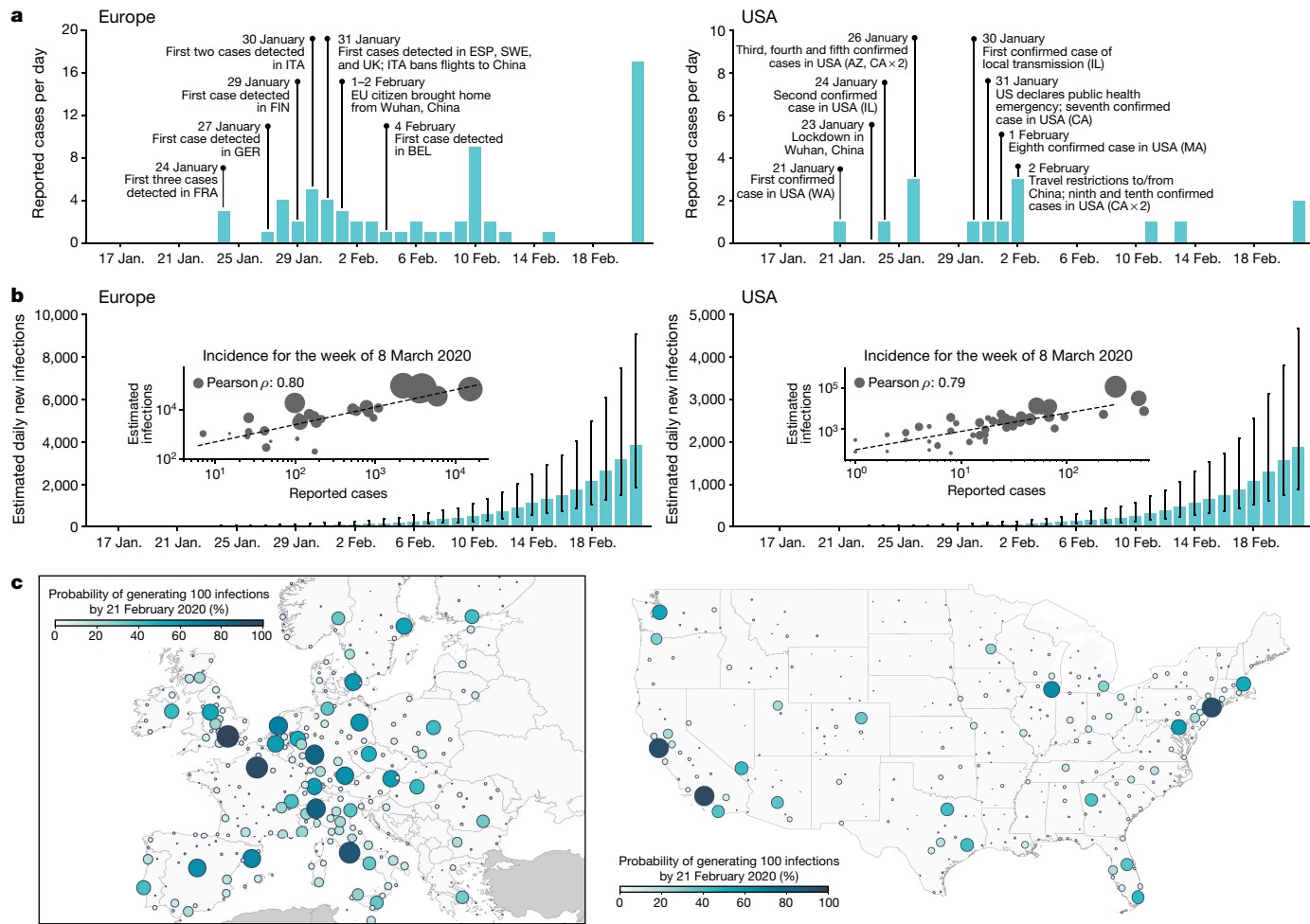

**Fig. 1 | Early picture of the COVID-19 outbreak in Europe and the USA.**
**a**, Timelines of the daily reported and confirmed cases of COVID-19 in Europe (left) and the USA (right). BEL, Belgium; ESP, Spain; EU, European Union; FIN, Finland; FRA, France; GER, Germany; ITA, Italy; SWE, Sweden. **b**, Model-based estimates for the daily number of new infections in Europe (left) and the USA (right). The model estimates reported are the median values with the IQR obtained with an ABC calibration method using $n = 200{,}000$ independent model realizations. The inset plots compare the weekly incidence of reported cases with the median, weekly incidence of infections estimated by the model

for the week of 8–14 March 2020 for the contiguous US states and European countries that reported at least one case (Europe, $n = 30$; USA, $n = 48$). Circle size corresponds to the population size of each state and country. The correlations were calculated using the Pearson correlation coefficient with a two-sided $P$ value (Europe: $\rho = 0.80$, $P < 0.001$; USA: $\rho = 0.79$, $P < 0.001$). **c**, The probability that a city in Europe (left) and the USA (right) had generated at least 100 cumulative infections by 21 February 2020. Colour and circle size are proportional to the probability.

including countries of world regions such as Latin America, the Middle East, Africa, East Asia and Oceania.

In Fig. 1b we show the model estimates of the median daily incidence of new infections up to 21 February 2020, for both the USA and Europe. These values are much larger than the number of officially reported cases (see Fig. 1a), highlighting the substantial number of potential transmission events that may have already occurred before many states and countries had implemented testing strategies independent of travel history. As validation we compare our model's estimates of the number of infections during the week of 8 March 2020 to the number of cases reported during that week within each US state and European country with at least one reported case (shown in Fig. 1b inset). While we see a strong correlation between the reported cases and our model's estimated number of infections (Pearson's correlation coefficient on log values, USA: 0.79, $P < 0.001$; Europe: 0.80, $P < 0.001$), far fewer cases had actually been reported by that time. If we assume that the number of reported cases and simulated infections are related through a simple binomial sampling process, we find that on average 9 in 1,000 infections (90% confidence interval (CI) 1–35 per 1,000) and 35 in 1,000 infections (90% CI 4–90 per 1,000) were detected by

8 March 2020 in the USA and Europe, respectively. As testing capacity increased, the ascertainment rate grows and our estimates increase to detecting 17 in 1,000 infections (90% CI 2–55 per 1,000) by 14 March 2020 in the USA and 77 in 1,000 infections (90% CI 5–166 per 1,000) in Europe. The estimated ascertainment rates are in agreement with independent results based on different statistical methodologies[14–16]. In Fig. 1c we show the probability that a city in the USA or Europe had generated at least 100 infections by 21 February 2020. We see that the progression of the virus through the USA and Europe was both temporally and spatially heterogeneous. While many cities had not yet experienced much community transmission by late February, a few areas such as New York City and London are very likely to have already had local outbreaks.

## Onset of local transmission

The model's ensemble of realizations provides a statistical description of all the potential pandemic histories compatible with the initial evolution of the pandemic in China. Rather than describing a specific, causal chain of events, we can estimate possible time windows pertaining to

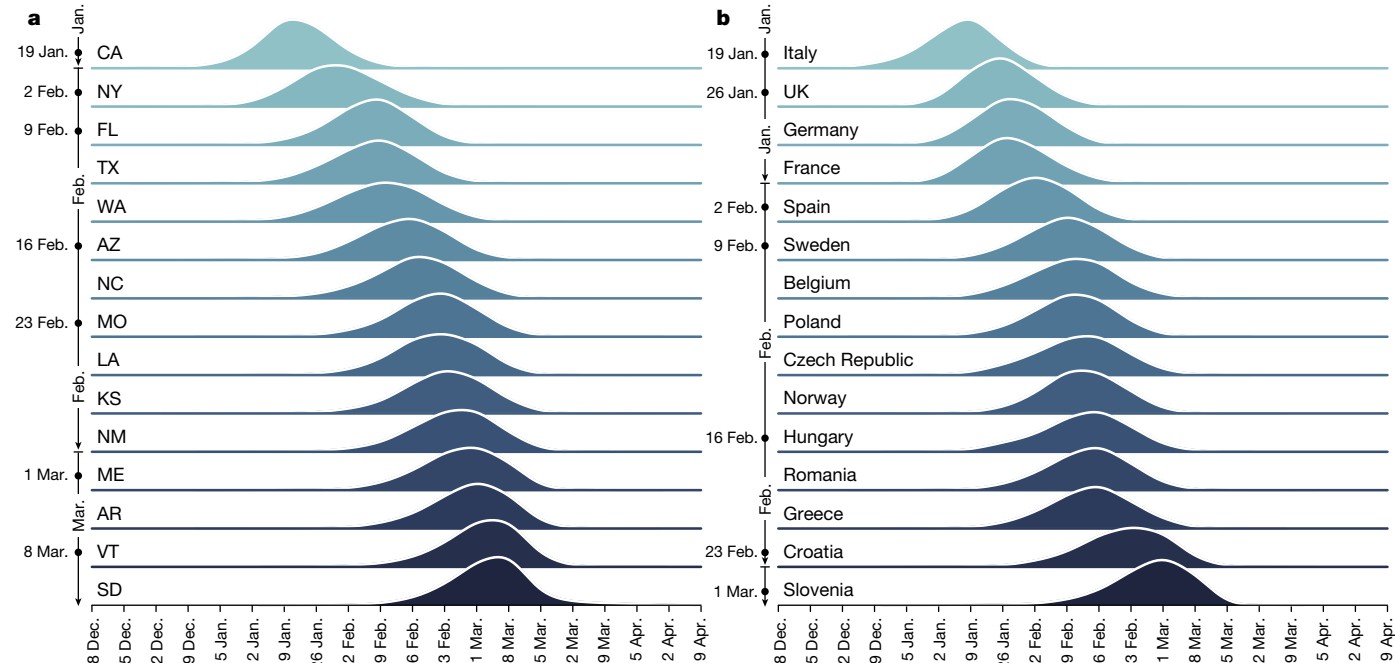

**Fig. 2 | Timing of the onset of local transmission. a**, **b**, Posterior distributions of the week in which each US state (**a**) or European country (**b**) first reached 10 locally generated SARS-CoV-2 transmission events per day.

Countries and states are ordered by the median date of their posterior distribution. The week of this date corresponds to the dates reported on the vertical axis.

the initial chains of transmission in different geographical regions. We define the onset of local transmission for a country or state as the earliest date when at least 10 new infections are generated per day. This number is chosen because at this threshold the likelihood of stochastic extinction is extremely small[17,18]. As detailed in the Supplementary Information, further calibration on the US states and European countries suggests posterior values of $R_0$ ranging from 2.4 to 2.8. These values are consistent with many other (country-dependent) estimates[19–24]. At the same time, given the doubling time of the number of COVID-19 cases before the implementation of public health measures, any variation of a factor 2 around the 10 infections per day threshold corresponds to a small adjustment of 3–5 days to the presented timelines.

In Fig. 2, we show the posterior probability distribution, $p(t)$, of the week, $t$, of the onset of local transmission for 15 US states (Fig. 2a) and European countries (Fig. 2b) (see Supplementary Information for all states and countries). We also calculate, for each country or state, the median date, $T$, that identifies the first week in which the cumulative distribution function is larger than 50%. Among the US states, CA and NY have the earliest dates, $T$, by the week of 19 January (CA) and 2 February (NY) 2020. In Europe, Italy, the UK, Germany and France are the first countries with $T$ close to the end of January 2020. However, it is worth noting that each distribution, $p(t)$, has a support spanning several weeks. In Italy, the 5th and 95th percentiles of the $p(t)$ distribution are the week of 6 January and the week of 30 January 2020, respectively. These dates also suggest that it is not possible to rule out introductions and transmission events as early as December 2019, although the probability of this is very small.

For each state in the USA and each country in Europe, we compared the order in which they surpassed 100 cumulative infections in the model and in the surveillance data (gathered from the John Hopkins University Coronavirus Resource Center[25]). In Extended Data Fig. 1a we plot the ordering for states and compute the Kendall rank correlation coefficient $\tau$ (see Supplementary Information). The correlation is positive ($\tau_{EU}$= 0.71, $P < 0.001$ and $\tau_{US}$ = 0.68, $P < 0.001$) indicating that, despite the detection and testing issues, the expected patterns of epidemic diffusion are largely described by the model in both regions.

## SARS-CoV-2 introductions

As the model allows the recording of the origin and destination of travellers carrying SARS-CoV-2 at the global scale, we can study the possible sources of SARS-CoV-2 introductions for each US state and European country. More specifically, we record the cumulative number of introductions in each stochastic realization of the model until 30 April 2020. In Fig. 3 we visualize the origin of the introductions considering some key geographical regions (for example, Europe and Asia) while keeping the USA and China separate and aggregating all of the other countries (Others). For both the USA and Europe, the contribution from mainland China is barely visible and the local share (that is, sources within Europe and the USA) becomes significantly higher across the board. Hence, while introduction events in the early phases of the outbreak were key to start local spreading (see details in the Supplementary Information), the cryptic transmission phase was sustained largely by internal flows. Domestic SARS-CoV-2 introductions to 30 April 2020 account for 69% (interquartile range (IQR) 60%–81%) of the introductions in CA, 78% (IQR 71%–87%) in TX and 69% (IQR 60%–80%) in MA, which is supported by phylogenetic analysis[26]. European origins account for 69% (IQR 60%–80%), 84% (IQR 79%–91%) and 58% (IQR 48%–68%) of the introductions in Italy, Spain and the UK, respectively. In the Supplementary Information, we report the full breakdown for all states and countries.

It is also necessary to distinguish between the full volume of SARS-CoV-2 introductions and the introduction events that could be relevant to the early onset of local transmission in each stochastic realization of the model. To this point, it is worth stressing that seeding introductions are different from the actual number of times the virus has been introduced to each location with subsequent onward transmission. Even after a local outbreak has started, future importation events may give rise to additional onward transmission forming independently introduced transmission lineages of the virus[27]. In the model, we can investigate seeding events by recording introduction events before the local transmission chains were established. We report the results of this analysis in the Supplementary Information, showing that importations from mainland China may be relevant in seeding the epidemic in January,

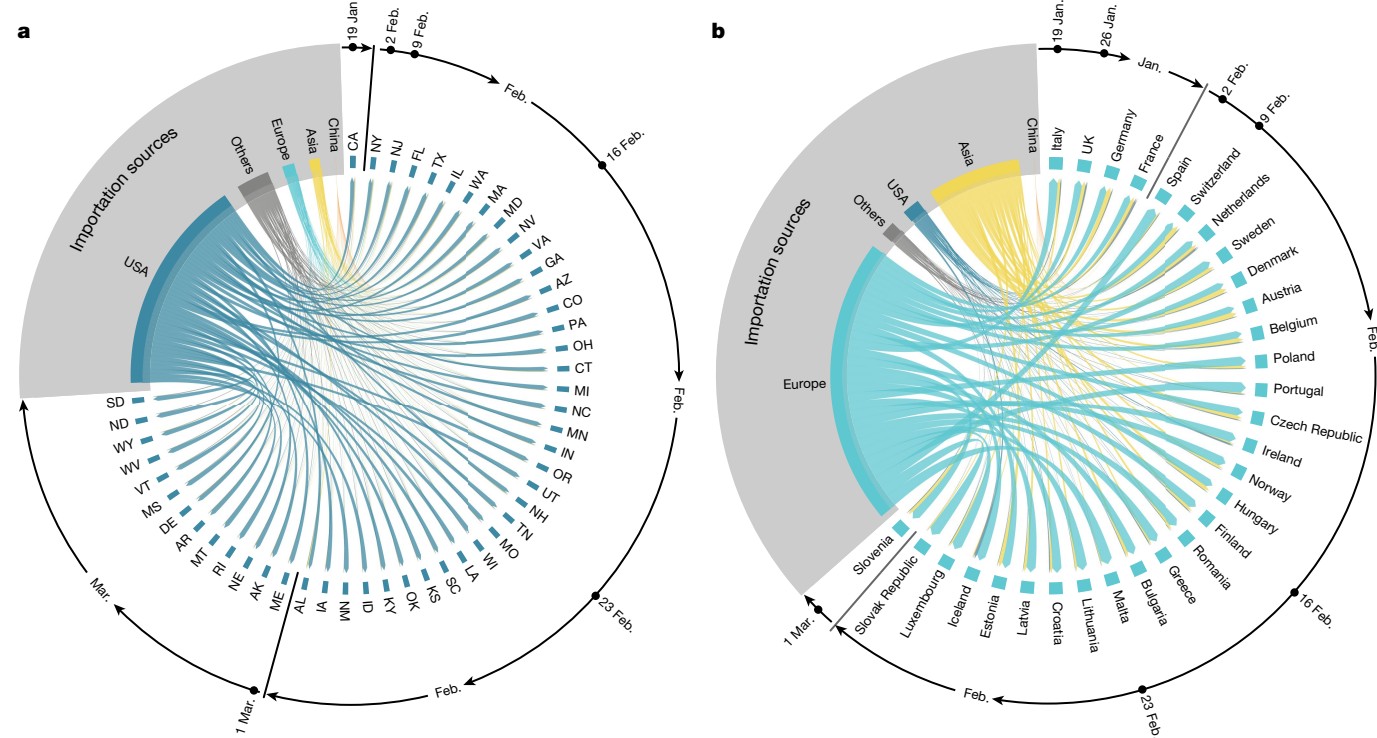

**Fig. 3 | Importation sources from the beginning of the outbreak until the end of April 2020. a, b**, Each US state (**a**) and European country (**b**) is displayed in a clockwise order with respect to the start of the local outbreak (as seen in Fig. 2). Importation flows are directed and weighted. We normalize links considering the total in-flow for each state so that the sum of importation flows, for each state, is 1. In the Supplementary Information, we report the complete list of countries contributing as importation sources in each geographical region.

but then play a comparatively small role in the expansion in the number of COVID-19 cases in the USA and Europe owing to the travel restrictions imposed to/from mainland China after 23 January 2020.

The early timing of the initial introductions and diffusion pattern of SARS-CoV-2 were driven by air travel. We find a positive correlation ($\tau_{EU} = 0.66$, $P < 0.001$ and $\tau_{US} = 0.66$, $P < 0.001$) comparing the ordering of states according to when they surpassed 100 cumulative, reported cases (referred to as the epidemic order) and their domestic and international air travel volume rank (Extended Data Fig. 1B). Similar observations have been reported in China, where the initial spreading of the virus outside Hubei was strongly correlated with the traffic to/from the province[28]. Other factors such as population size are also correlated with both the travel flows ($\tau_{EU} = 0.59$, $P < 0.001$ and $\tau_{US} = 0.7$, $P < 0.001$) and the epidemic order ($\tau_{EU} = 0.46$, $P < 0.001$ and $\tau_{US} = 0.68$, $P < 0.001$), which are discussed in detail in the Supplementary Information. In our model, it is not possible to exclude increased contacts in highly populated places before social distancing interventions and disentangle this effect from increased seeding due to the correlation between travel volume and population size.

## COVID-19 burden

Starting in March 2020, the establishment and timing of NPIs as well as other epidemiological drivers (that is, population size and density, age structure and so on) determined the disease burden in the USA and Europe[29–32]. We account for these features by calibrating the model results, individually, for each US state and European country. More precisely, we estimate the posterior distribution of the infection fatality ratio (IFR) and infection attack rate in each US state and European country. To this end, we adopt the ABC approach using as evidence the number of new deaths reported from 22 March 2020 to 27 June 2020. We consider a uniform prior for the average IFR in the range from 0.4% to 2% that is age stratified proportional to the IFR values reported in

ref.[33]. We also consider a uniform prior for reporting delays between the date of death and reporting ranging from 2 to 22 days in both Europe and the USA[34]. Details are provided in the Supplementary Information.

In Fig. 4a–d, f–i, we report the model fit of the estimated weekly deaths of the first wave for selected states and countries. Additional model results for all investigated regions including a sensitivity analysis of different calibration methods can be found in the Supplementary Information. We find a strong correlation between the weekly model-estimated deaths and the reported values with a Pearson correlation coefficient of 0.99 ($P < 0.001$) for both Europe and the USA (see Supplementary Fig. 6). As the data suggest, many European countries and US states saw peaks in April and May with various decreasing trajectories that depend on the mitigation strategies in place. Additionally, we report the estimated posteriors for the cumulative infection attack rates and IFRs as of 4 July 2020 in European countries experiencing more than 100 total deaths and the top 20 states ranked by infection attack rate in the USA.

Within Europe, Belgium has the highest estimated infection attack rate of 13.2% (90% CI [8.5%–28.3%]) by 4 July 2020, in agreement with the results in ref.[14]. Furthermore, by that time Belgium reported the highest COVID-19 mortality rate out of the European countries investigated with 8.5 deaths per 10,000 individuals. However, Italy is estimated to have the highest median IFR of 1.4% (90% CI [0.6%–1.8%]), which aligns with other ranges reported in the literature[35,36]. The US states with the highest infection attack rates are located within the northeast and experienced a significant first wave during March–April 2020. NY and NJ are the top two states with infection attack rates of 13.4% (90% CI [9.1%–26.7%]) and 15.2% (90% CI [10.2%–31.3%]), respectively. These numbers are aligned with estimates from New York City reported in ref.[37]. In the Supplementary Information, we report summary tables with estimated IFRs, infection attack rates and the reproductive number in the absence of mitigation measures for all calibrated US states and European countries. Additionally, we compare our attack rate

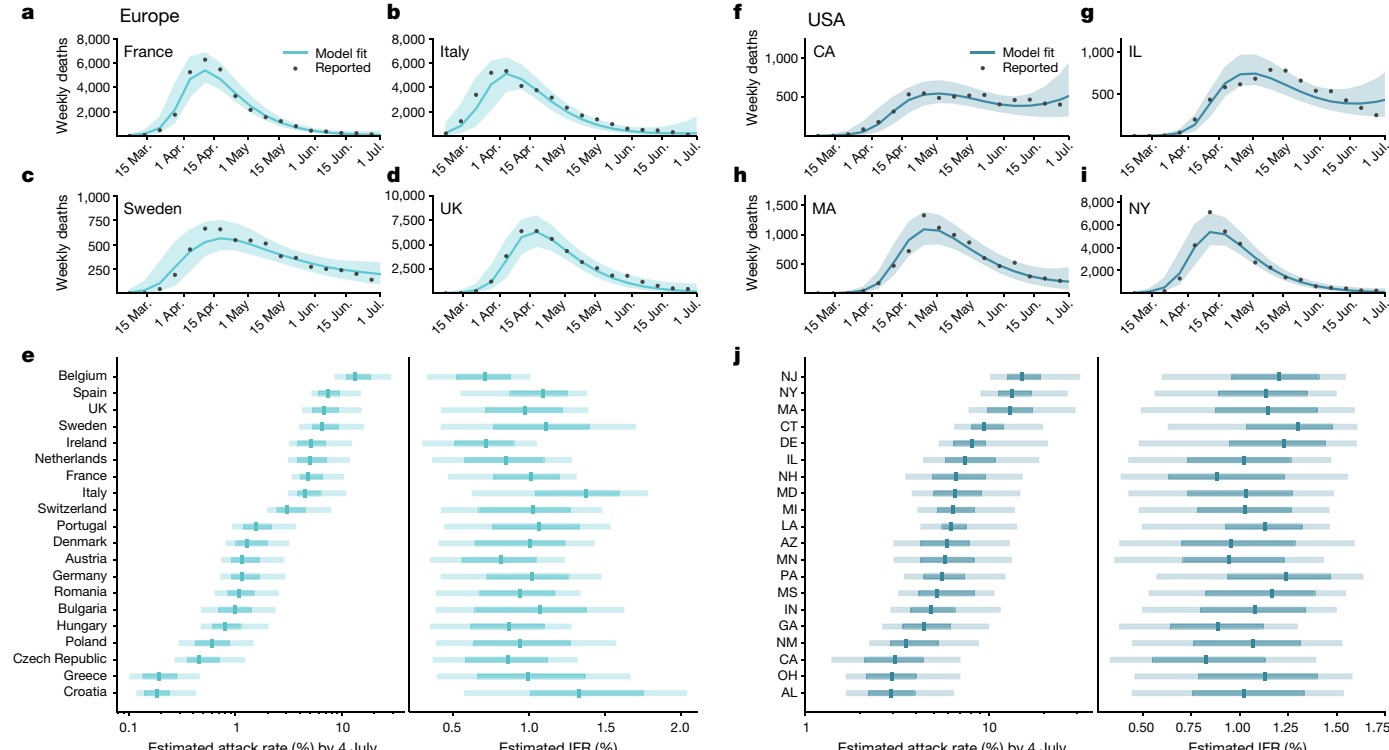

**Fig. 4 | The burden of the first wave in Europe and the USA. a–d**, Model fit of the estimated weekly deaths for selected countries in Europe (France, **a**; Italy, **b**; Sweden, **c**; UK, **d**). **e**, Posterior distributions of the infection attack rates and IFRs by 4 July 2020, for European countries where there were at least 100 reported deaths. **f–i**, Model fit of the estimated weekly deaths for selected states in the USA (CA, **f**; IL, **g**; MA, **h**; NY, **i**). **j**, Posterior distributions of the estimated infection attack rates and IFRs by 4 July 2020 for the top 20 US states (ranked according to their infection attack rates). The curves in **a–d** and **f–i** show the median values and 90% CIs. For **e** and **j**, the outer, lighter boxes represent the 90% CI, the darker, inner boxes represent the IQR, and the vertical lines represent the median value. Posterior distributions in **e** and **j** are the result of the ABC analysis of 200,000 independent model realizations.

estimates to the prevalence of individuals with SARS-CoV-2 antibodies from serological studies across the USA and Europe (Extended Data Fig. 1D). The seroprevalence estimates are compared to the model estimates during the same time window the studies were performed (details on the seroprevalence data from this figure can be found in Supplementary Table 8 and Supplementary Section 9.3).

## Discussion

The model presented here captures the spatial and temporal heterogeneity of the early stage of the pandemic, going beyond the single-country-level reconstruction. It provides a mechanistic understanding of the underlying dynamics of the pandemic's interconnected evolution. Furthermore, rather than showing specific evidence for early infection in a few locations, our study aims at providing a statistical characterization and quantification of the initial transmission pathways at a global scale. Our results can be compared to and complement analyses based on gene sequencing and travel volumes. We find that 72% of the early introductions to Italy, before the local outbreak, are linked to China, which is in agreement with ref.[38] highlighting the key role of importations between these regions at the beginning of the pandemic. Additionally, similar to our findings, ref.[27] estimates that the majority of importation events to April 2020, associated with onward transmission in the UK, came from Europe. The contributions from China are quantified below 1% and limited to the very early phase. Furthermore, seeding events from the USA are estimated to be below 3%, which aligns with our estimate (8%; IQR 3%–9%). However, their results point to a larger share from Europe (~90%) compared to ours (58%; IQR 48%–68%), and conversely, we estimate a larger contribution from Asia (27%; IQR 19%–35%). As our analysis is a statistical description of the possible introduction pathways, differences could arise due to our model design, and also from genomic sampling biases[39].

The sources of introduction of SARS-CoV-2 infections in Europe and the USA changed substantially and rapidly through time. This caused reactive response strategies, such as issuing travel restrictions targeting countries only after local transmission is confirmed, ineffective at preventing local outbreaks. Our results suggest that many regions in the USA and Europe experienced an onset of local transmission in January and February 2020, during the time when testing capacity was limited. If testing had been more widespread and not restricted to individuals with a travel history from China, there would have been more opportunities for earlier detection and interventions. In the Supplementary Information, we report a counterfactual scenario where we assume broader testing specifications not based on the individual travel history and find that the epidemic progression is considerably delayed (see Supplementary Section 8).

As testing capacity increased and more cases were detected, many governments began to issue social distancing guidelines to mitigate the spread of SARS-CoV-2. The first European country to implement a cordon sanitaire was Italy on 23 February 2020, for a few northern cities[40]. Many other countries followed suit and implemented national lockdowns in March 2020 (refs.[30,41]); however, this was weeks after our model estimates that SARS-CoV-2 was introduced and locally spreading. We find a strong correlation between the number of cases reported by the date of a lockdown/social distancing measure and the cumulative infections projected by 4 July 2020 (Extended Data Fig. 1C), indicating that the earlier NPIs had been issued, the smaller the COVID-19 burden experienced during the first wave. This is in agreement with other analyses showing that the timing of NPIs is crucial in limiting the burden of COVID-19 (refs.[19,29,42–48]). Overall, our results strengthen the case for preparedness plans with broader indication for testing that are able to detect local transmission earlier.

As with all modelling analyses, results are subject to biases from the limitations and assumptions within the model as well as the data used in its

calibration. The model's parameters, such as generation time, incubation period and the proportion of asymptomatic infections, are chosen according to the current knowledge of SARS-CoV-2. Although the model is robust to variations in these parameters (see the Supplementary Information for the sensitivity analysis), more information on the key characteristics of the disease would considerably reduce uncertainties. The model calibration does not consider correlations among importations (that is, family travel) and assumes that travel probabilities are age specific across all individuals in the catchment area of each transportation hub.

In light of the assumptions and limitations inherent to this modelling approach, the results are able to complement the SARS-CoV-2 genome sequencing analyses used to reconstruct the early epidemic history of the COVID-19 pandemic[38]. The methods used in this analysis offer a blueprint to identify the most likely early spreading dynamics of emerging viruses, and they can be used as a real-time risk assessment tool. Anticipating the locations where a virus is most likely to spread to next could be instrumental in guiding enhanced testing and surveillance activities. The estimated SARS-CoV-2 importation patterns and the cryptic transmission phase dynamics are of potential use when planning and developing public health policies in relation to international travelling, and they could provide important insights into assessing the potential risk and impact of emerging SARS-CoV-2 variants in regions of the world with limited testing and genomic surveillance resources.

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

# Methods

## The GLEAM model

The GLEAM model is a stochastic, spatial, age-structured metapopulation model. Previously this model was used to characterize the early stage of the COVID-19 epidemic in mainland China to estimate the effectiveness of travel bans and restrictions[49]. The GLEAM model divides the global population into more than 3,200 subpopulations in roughly 200 different countries and territories interconnected by realistic air-travel and commuting mobility networks. A subpopulation is defined as the catchment area around major transportation hubs. The airline transportation data encompass daily travel data in the origin–destination format from the Official Aviation Guide database[50] reflecting actual traffic changes that occurred during the pandemic. Ground mobility and commuting flows are derived from the analysis and modelling of data collected from the statistics offices of 30 countries on 5 continents[11,12]. The international travel data account for travel restrictions and government-issued policies. Furthermore, the model accounts for the reduction of internal, country-wide mobility and changes in contact patterns in each country and state in 2020. Specific model details are reported in the Supplementary Information.

## SARS-CoV-2 transmission dynamics

The transmission dynamics take place within each subpopulation and assume a classic compartmentalization scheme for disease progression similar to those used in several large-scale models of SARS-CoV-2 transmission[15,51–55]. Each individual, at any given point in time, is assigned to a compartment corresponding to their particular disease-related state (specifically, one could be susceptible, latent, infectious or removed)[49]. This state also controls the individual's ability to travel (details in the Supplementary Information). Individuals transition between compartments through stochastic chain binomial processes. Susceptible individuals can acquire the virus through contact with individuals in the infectious category and can subsequently become latent (that is, infected but not yet able to transmit the infection). The process of infection is modelled using age-stratified contact patterns at the state and country level[56,57]. Latent individuals progress to the infectious stage at a rate inversely proportional to the latent period, and infectious individuals progress to the removed stage at a rate inversely proportional to the infectious period. The sum of the mean latent and infectious periods defines the generation time. Removed individuals are those who can no longer infect others. To estimate the number of deaths, we consider a uniformly distributed prior of the IFRs (ranging from 0.4% to 2%) that is age stratified proportional to the values estimated by ref.[33] and incorporates reporting delays. The transmission model does not assume heterogeneities due to age differences in susceptibility to the SARS-CoV-2 infection for younger children (1–10 years old). This is an intense area of discussion[58,59,60]. The transmission dynamic and the offspring distribution of infectious individuals in the model will depend on the specific details of each population, local and global mobility, NPIs and so on. While overdispersion in transmission varies by location in our model, we find that overall, it is consistent with 25% of primary infections causing 75% of transmission in our simulations (Supplementary Fig. 9). Additional simulations considering a fixed level of dispersion, informed by past studies, result in differences of less than 3 days in onset times (Supplementary Fig. 10; see also the Supplementary Information for further discussion).

## Model calibration

We assume a start date of the epidemic in Wuhan, China, that falls between 15 November 2019 and 1 December 2019, with 20 initial infections[49,51,61,62,63]. This considers that our model has a posterior distribution for the emergence of the outbreak in China that includes the possibility of transmission starting in October, 2019 (refs.[64,65]). The model generates an ensemble of possible epidemic realizations and is initially calibrated using an approximate Bayesian computation (ABC) rejection approach[13] based on the observed international importations from mainland China up to 21 January 2020 (ref.[49]). Only a fraction of imported cases is generally detected at the destination[10,66]. According to the estimates proposed in ref.[67], we stratify the detection capacity of countries into three groups: high, medium, and low surveillance capacity according to the Global Health Security Index[68], and assume that asymptomatic infections are never detected. The model calibration does not consider correlated importations (for example, family travel) and assumes that travel probabilities are homogeneous across all individuals in each subpopulation. We further calibrate our model using the temporal ordering of the onset of local transmission (as defined in the section 'Onset of local transmission') of the countries investigated. If we consider the epidemiological evidence[38,69,70], Italy was the first European country to experience substantial community transmission. Therefore, throughout the paper, we constrain the ensemble of simulations focusing only on stochastic realizations in which Italy is the first country, in the group under examination, to experience sustained local transmission (see Supplementary Information for details and further analyses of unconstrained simulations). Furthermore, we perform for each state and country an additional ABC rejection analysis using as evidence the weekly reported deaths in the time window starting on 22 March 2020 and ending on 27 June 2020. A full description of the model calibration is provided in the Supplementary Information.

## Reporting summary

Further information on research design is available in the Nature Research Reporting Summary linked to this paper.

## Data availability

Epidemic surveillance data were collected from the Johns Hopkins Coronavirus Resource Center (https://coronavirus.jhu.edu/). Proprietary airline data are commercially available from the Official Aviation Guide (https://www.oag.com/) and International Air Transport Association (https://www.iata.org/) databases. Other model intervention data include data from Google's COVID-19 Community Mobility Reports available at https://www.google.com/covid19/mobility/ and the Oxford COVID-19 Response Tracker available at https://github.com/OxCGRT/covid-policy-tracker. Source data are provided with this paper.

## Code availability

The GLEAM model is publicly available at http://www.gleamviz.org/. All data analyses of model results were performed using Python v3.8.

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

**Acknowledgements** A.V., M.E.H., N.E.D. and I.M.L. acknowledge support from the award NIH-R56AI148284. S.M. acknowledges support from the EU H2020 MOOD project. C.G. and L.R. acknowledge support from the EU H2020 Icarus project. M.A., M.C. and A.V. acknowledge support from the COVID Supplement CDC-HHS-6U01IP001137-01. M.C. and A.V. acknowledge support from the Google Cloud Research Credits programme to fund this project. A.V. acknowledges support from the McGovern Foundation and the Chleck Foundation. The findings and conclusions in this study are those of the authors and do not necessarily represent the official position of the funding agencies, the National Institutes of Health or the US Department of Health and Human Services.

**Author contributions** J.T.D., M.C., N.P. and A.V. designed research; M.C., J.T.D., N.P., M.A., C.G., M.L., S.M., A.P.P., K.M., L.R., K.S., C.V., X.X., M.E.H., I.M.L. and A.V. performed research; M.C., J.T.D., N.P., A.P.P., K.M. and A.V. analysed data; and M.C., J.T.D., N.P., M.A., C.G., M.L., S.M., A.P.P., K.M., N.E.D., L.R., K.S., C.V., X.X., M.E.H., I.M.L. and A.V. wrote and edited the paper.

**Competing interests** M.A. reports research funding from Seqirus, not related to COVID-19. A.V., M.C. and A.P.P. report grants from Metabiota Inc., outside the submitted work. The authors declare no other relationships or activities that could appear to have influenced the submitted work.

**Additional information**
**Correspondence and requests for materials** should be addressed to Alessandro Vespignani.

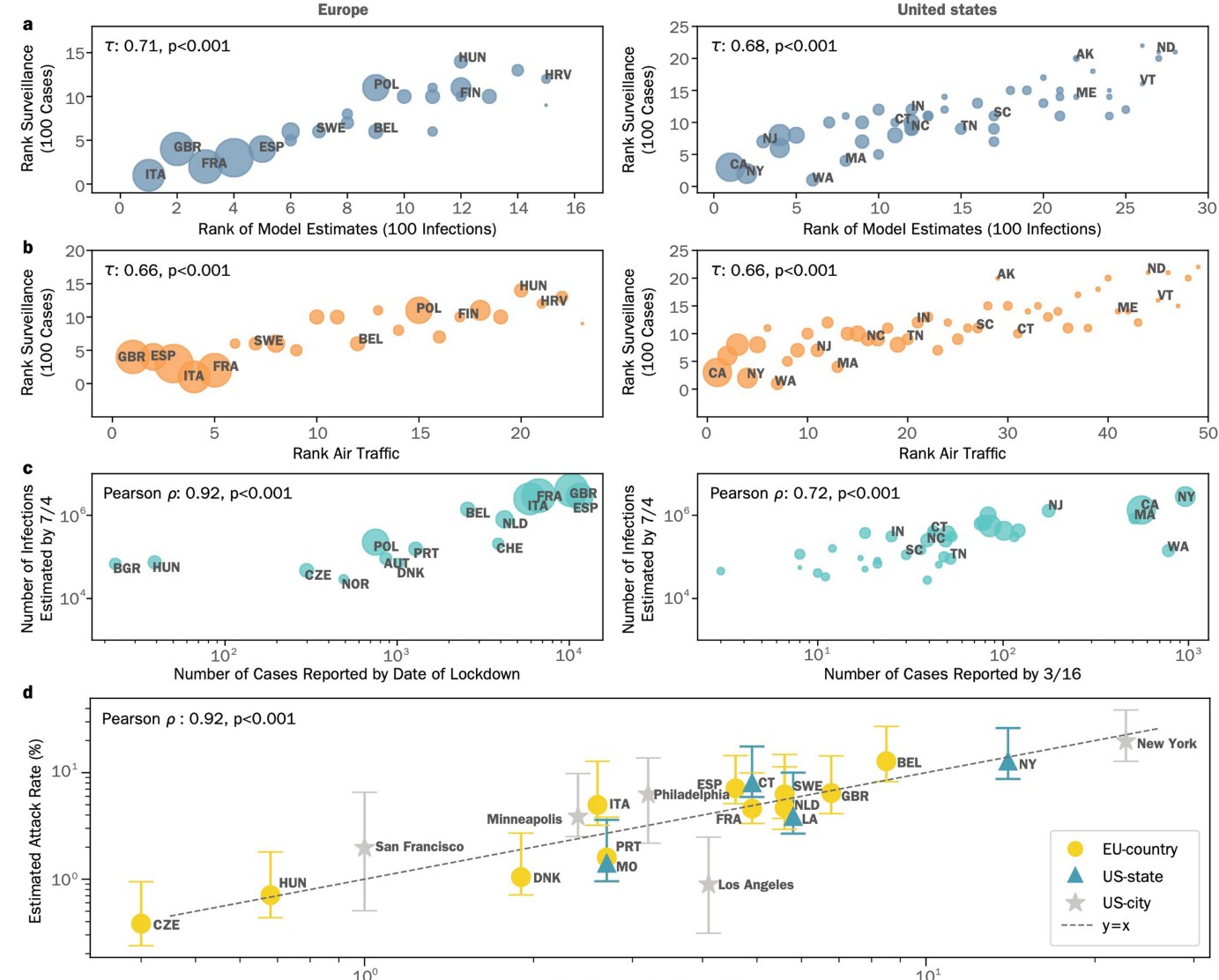

**Extended Data Fig. 1 | Correlation Analysis for European countries and US states.** (**a**) The correlation between the ordering of each country/state to reach 100 infections in the model-estimates and to reach 100 reported cases in the surveillance data (Europe: n = 23, US: n = 49). (**b**) The correlation between the ordering of each country/state considering the time needed to reach 100 reported cases in the surveillance data and the ranking of the combined international and domestic air traffic (Europe n = 23, US n = 49). Correlations in (**a**, **b**) are computed considering the Kendall rank correlation coefficient reported with a two-sided p-value, we consider European countries that reached at least 100 reported deaths by July 4, 2020 and countries in Scandinavia (**c**) Left: the correlation between the number of cases reported by the date of lockdown for European countries (from Table 4 in Ref.[71]) and the estimated total number of infections by July 4, 2020 (median values, n = 15). Right: the correlation between the number of cases reported by March 16, 2020

(the date the "15 days to slow the spread" guidelines were released in the US Ref.[72]) for each US state and the estimated total infections by July 4, 2020 (median values, n = 36). We consider states that reached at least 100 reported deaths by July 4, 2020. The circle sizes in (**a**–**c**) correspond to the population sizes of each country/state. (**d**) The correlation between the model-estimated infection attack rate and the serological prevalence collected from studies, n = 20. Estimated attack rates are the posterior distributions that are the result of the ABC analysis of 200,000 independent model realizations. Data points refer to different dates and the locations for which serological surveys were available (see table S8 in SI for study descriptions). The model-estimated attack rates use the median value, and the error bars represent the 90%CI. The correlations are calculated using the Pearson correlation coefficient in (**c**, **d**) reported with a two-sided p-value.

**Extended Data Table 1 | Regions under investigation**

| United States | Europe |
|---|---|
| Alabama, Alaska, Arizona, Arkansas, California, Colorado, Connecticut, Delaware, Florida, Georgia, Idaho, Illinois, Indiana, Iowa, Kansas, Kentucky, Louisiana, Maine, Maryland, Massachusetts, Michigan, Minnesota, Mississippi, Missouri, Montana, Nebraska, Nevada, New Hampshire, New Jersey, New Mexico, New York, North Carolina, North Dakota, Ohio, Oklahoma, Oregon, Pennsylvania, Rhode Island, South Carolina, South Dakota, Tennessee, Texas, Utah, Vermont, Virginia, Washington, West Virginia, Wisconsin, Wyoming | Austria, Belgium, Bulgaria, Croatia, Czech Republic, Denmark, Estonia, Finland, France, Germany, Greece, Hungary, Iceland, Ireland, Italy, Latvia, Lithuania, Luxembourg, Malta, Netherlands, Norway, Poland, Portugal, Romania, Slovak Republic, Slovenia, Spain, Sweden, Switzerland, United Kingdom |

List of European countries and US states analyzed.

# Reporting Summary

## Statistics

For all statistical analyses, confirm that the following items are present in the figure legend, table legend, main text, or Methods section.

| n/a | Confirmed | |
|---|---|---|
| ☒ | ☐ | The exact sample size (*n*) for each experimental group/condition, given as a discrete number and unit of measurement |
| ☒ | ☐ | A statement on whether measurements were taken from distinct samples or whether the same sample was measured repeatedly |
| ☐ | ☒ | The statistical test(s) used AND whether they are one- or two-sided *Only common tests should be described solely by name; describe more complex techniques in the Methods section.* |
| ☒ | ☐ | A description of all covariates tested |
| ☒ | ☐ | A description of any assumptions or corrections, such as tests of normality and adjustment for multiple comparisons |
| ☐ | ☒ | A full description of the statistical parameters including central tendency (e.g. means) or other basic estimates (e.g. regression coefficient) AND variation (e.g. standard deviation) or associated estimates of uncertainty (e.g. confidence intervals) |
| ☐ | ☒ | For null hypothesis testing, the test statistic (e.g. *F*, *t*, *r*) with confidence intervals, effect sizes, degrees of freedom and *P* value noted *Give P values as exact values whenever suitable.* |
| ☐ | ☒ | For Bayesian analysis, information on the choice of priors and Markov chain Monte Carlo settings |
| ☒ | ☐ | For hierarchical and complex designs, identification of the appropriate level for tests and full reporting of outcomes |
| ☒ | ☐ | Estimates of effect sizes (e.g. Cohen's *d*, Pearson's *r*), indicating how they were calculated |

*Our web collection on statistics for biologists contains articles on many of the points above.*

## Software and code

Policy information about availability of computer code

| Data collection | No software was used for data collection |
|---|---|
| Data analysis | The GLEAM model is publicly available at http://www.gleamviz.org/. All data analyses of model results were performed using python v3.8 |

For manuscripts utilizing custom algorithms or software that are central to the research but not yet described in published literature, software must be made available to editors and reviewers. We strongly encourage code deposition in a community repository (e.g. GitHub). See the Nature Portfolio guidelines for submitting code & software for further information.

## Data

Policy information about availability of data

All manuscripts must include a data availability statement. This statement should provide the following information, where applicable:
- Accession codes, unique identifiers, or web links for publicly available datasets
- A description of any restrictions on data availability
- For clinical datasets or third party data, please ensure that the statement adheres to our policy

Epidemic surveillance data were collected from the Johns Hopkins Coronavirus Resource Center https://coronavirus.jhu.edu/. Proprietary airline data are commercially available from OAG (https://www.oag.com/) and IATA (https://www.iata.org/) databases. Other model intervention data includes Google's COVID-19 Community Mobility Reports available at https://www.google.com/covid19/mobility/ and the Oxford COVID-19 Response Tracker available at https://github.com/OxCGRT/covid-policy-tracker.

# Field-specific reporting

Please select the one below that is the best fit for your research. If you are not sure, read the appropriate sections before making your selection.

☒ Life sciences　　　☐ Behavioural & social sciences　　　☐ Ecological, evolutionary & environmental sciences

For a reference copy of the document with all sections, see nature.com/documents/nr-reporting-summary-flat.pdf

# Life sciences study design

All studies must disclose on these points even when the disclosure is negative.

| | |
|---|---|
| Sample size | We use all available data generated by model simulations. We do not generate primary biological or epidemiological data from field experiments. |
| Data exclusions | No data were excluded |
| Replication | All data used are described in the data availability statement. Model generated data were generated synthetically using the GLEAM tool documented here: http://www.gleamviz.org/simulator/GLEAMviz_client_manual_v7.0.pdf |
| Randomization | N/A. We did not perform/consider individual subject studies.We did not allocate any individuals to control or experimental groups. |
| Blinding | N/A. We did not perform/consider individual subject studies. We did not allocate any individuals to control or experimental groups. |

# Reporting for specific materials, systems and methods

We require information from authors about some types of materials, experimental systems and methods used in many studies. Here, indicate whether each material, system or method listed is relevant to your study. If you are not sure if a list item applies to your research, read the appropriate section before selecting a response.

## Materials & experimental systems

| n/a | Involved in the study |
|---|---|
| ☒ ☐ | Antibodies |
| ☒ ☐ | Eukaryotic cell lines |
| ☒ ☐ | Palaeontology and archaeology |
| ☒ ☐ | Animals and other organisms |
| ☒ ☐ | Human research participants |
| ☒ ☐ | Clinical data |
| ☒ ☐ | Dual use research of concern |

## Methods

| n/a | Involved in the study |
|---|---|
| ☒ ☐ | ChIP-seq |
| ☒ ☐ | Flow cytometry |
| ☒ ☐ | MRI-based neuroimaging |

