## [Peer Review File · Nature]

Manuscript Title: Cryptic transmission of SARS-CoV-2 and the first COVID-19 wave

Reviewer Comments & Author Rebuttals

Reviewer Reports on the Initial Version:

Referee #1 (Remarks to the Author):

In this paper, authors use a metapopulation model to provide a mechanistic understanding of the global dynamic underlying the establishment of the COVID-19 pandemic in Europe and the US. Authors find that widespread community transmission of SARS-CoV-2 was likely in several areas of Europe and the US by January 2020 and estimate low detection rates in early March 2020. Authors characterize heterogeneity in spatio-temporal spread as well as the burden of the first COVID-19 wave.

I find the paper interesting but in its current form, there is a relative lack of novelty. I believe authors might be able to do more to draw conclusions/lessons learnt from their work.

Major comment:

1) Explaining 1 pandemic realization versus drawing general conclusions about the international spread of pandemics: I see three major limitations of the current manuscript

- a. The key finding highlighted by the paper is that there was cryptic transmission in Europe and the US in the early stage of the pandemic. This is interesting but has already been documented in numerous studies; so this is not really a novel finding.
- b. The reconstruction of the trajectory of the pandemic as it happened is another finding highlighted in multiple figures of the paper. However, this is done with a model that is not calibrated to the whole data available but only to information about which countries where the first impacted. This puts the authors in an awkward position to explain the discrepancies with the genetic data for example "This discrepancies might be due to ... the fact that we sample all possible epidemics paths statistically possible rather than the single, observed occurrence". If the aim is really to reconstruct for each location the time of onset of local transmission in state X or country Y (as is for example presented in Figure 3), it is hard to justify performing an analysis of all possible paths rather than the single observed one. I would say this is a major limitation of the paper in its current form.
- c. Insights gained about control strategies for future pandemics remain relatively limited. Authors write: "reactive response strategies, such as issuing travel restrictions targeting countries only after local transmission is confirmed are highly inefficient." However they have not used their model to test and compare different control strategy.

So, while the current work is interesting, I do not think it makes best use of the metapopulation model that is being presented. The strength of this model is not to reconstruct the single observed pandemic realization but to explore many possible scenarios of international spread with different types of restrictions on international travel. In this context, I think that the paper would be substantially strengthened if it was reframed with the aim to draw general conclusions about international spread for future pandemics (and less emphasise on reconstructing what happened). After authors have demonstrated that the model fitted the current realization relatively well and documented cryptic transmission, they could use their model to explore how pandemic spread might have changed if things had been done differently. For example, they claim that strategies targeting countries only after local transmission is confirmed are inefficient. Knowing what they know now, what could have been gained by implementing earlier more stringent travel restrictions? What would be their recommendations for testing and travel restrictions next time the world is confronted to this kind of situation? If the major reductions in air transportation had occurred 1 or 2 months early, how would that have impacted international spread?

I felt that the last part of the paper estimating Infection Fatality Ratios was disconnected from the

rest of the paper (in this part there is additional calibration to more country data and it considers a longer time period than the early stage of cryptic transmission). My suggestion would therefore be to refocus the paper on international spread and implications for future pandemics, remove this part and add simulation studies of interventions at international level.

Minor comments:

Figure 1B: since deaths is a better proxy of SARS-CoV-2 circulation than cases, can authors compare observed and predicted number of deaths by location.

Line 154: "The wide distribution of dates suggests that SARS-CoV-2 cryptic transmission may have begun as early as December 2019." From Figure 2, it does not look like there was high likelihood of cryptic transmission in December. What is the associated probability?

Figure 2: Isn't that redundant with Figure 1C? If authors feel it's not, then I think a map of that graph would be more informative so we can understand spatial patterns (for example, colour the country/state as a function of the date of onset).

Figure 3: Authors need to clarify in the legend over which time period the importation source is characterized. Overall, I do not find that the Figure 3 in the main text is very informative for the discussion about the early stage of the pandemic as it seems quite obvious that if we consider an extended period up to the end of April, SARS-CoV-2 will be everywhere and so most introductions will be from neighbouring areas. I feel that the equivalent of Figure 3 that is in the Supplement and shows where introductions come from at the start of local transmission, is actually much more interesting. For Europe for example, we see a clear signal where first impacted countries were infected by China, then Asian countries and then European countries. If I was to select one of these two figures, my preference would clearly go to Figure S9 (arguably, the 2 would fit in the main manuscript – but I don't find current Fig 3 that covers a much wider time period very informative – as it moves away from the period of cryptic transmission that is the focus of the paper). In terms of presentation: i) Since there are only 5 possible origins/colours, I feel the current presentation is overly complex. Results would be clearer if they were represented with a barplot, with data for each state/country being presented with a single colored bar (1 color showing the proportion of the associated origin), and with the bars ordered by the date of introduction. ii) for colour coding, I think it would be clearer if the same colours were used for the same locations in the 2 panels (currently, Europe and the US switch colours between figures which I find confusing).

Concerning the air transportation data: if an individual goes from Asia to Italy with a transit in France, will they be counted as an importation from France? This needs to be clarified as this would represent an important limitation to the analysis.

Page 7, line 318: "reactive response strategies, such as issuing travel restrictions targeting countries only after local transmission is confirmed are highly inefficient." I do not think authors have demonstrated this point (that is, they haven't presented an evaluation of this specific aspect). First, the start of local transmission is defined by authors with a relatively small number of local infections. It is not clear that for such low numbers, the risk of exportation to other countries is important; so, it's perfectly possible that waiting a bit to implement travel restrictions is actually ok. Second, it may just be that travel restrictions are not very effective at limiting the risk of importation.

Methods, line 357: "To estimate the number of deaths we use as prior the age-stratified infection fatality ratios from Ref 48." The use of a prior would make sense if authors were trying to estimate the infection fatality ratio knowing the number of deaths. Here, they write that the aim is to estimate the number of deaths. So, are the estimates from ref 48 a prior? Or do they assume IFRs equal to those in ref 48? This needs to be clarified.

Does Figure 6 in the supplement show the correlation observed-expected number of deaths with the ABC step considering deaths data in individual countries (as the one used in Figure 4)? Or does it show the unconstrained simulations?

Figure 4-6 in the supplement: "using the calibration reported in the main text". It's not clear which calibration authors are talking about since results for 2 types of calibrations are used in the main text of the article.

Please label differently Figures in the Supplement (e.g. S1, S2...) and in the main text to avoid confusion.

In section 3.2 in the supplement, authors present an "Alternative distance measure for model calibration" where the ABC is also calibrated on deaths data. Is this similar to the method they used to estimate IFRs?

Referee #2 (Remarks to the Author):

This is an important study of the early transmission dynamics of COVID-19 and will be a seminal contribution to the literature when published. I think it is important to distinguish what /could have happened/ (from simulations) with /what actually happened/, which I think authors have done a good job of this. Since you have a global model, why focus on US and Europe but not considering other locations in Asia, or Africa and South America? Iran was also very heavily affected in the early days, if I remember correctly.

Minor comments:

Abstract - Infection attack rates of 0.78% to 15.2% and 0.19% to 13.2% - these are very wide ranges, could you name some of the locations with estimated higher or lower incidence to make it clearer the variability is in location-specific estimates?

In line 136-7, you mention the extinction probability, does this take into account overdispersion (superspreading)?

Referee #3 (Remarks to the Author):

Overall

There authors present analysis from a well-known and validated global model of infectious disease transmission. They address the question of how quickly the virus spread initially and how much earlier many populations were seeded than was evident in the public case and death data. They conclude that there was likely widespread transmission in many populations much earlier than is currently assumed. This would be a major claim if published here.

Some key results are compared with those from the coalescent model that is used by evolutionary virologists to address the same question. Where the models disagree, the authors point to potential biases in the coalescent work. Although neither approach is free from bias, the evolutionary models make use of much more data in combining dates and locations of whole genomes. The epi model here can be thought of as a forward simulation of the linked exponential processes assumed when fitting the phylogeographic models (or looking at the tip locations in non-geographic phylogenetic trees). Also, it could be argued that variable growth rates in the coalescent model are more flexible for this type of question.

The underlying process of global spread and establishment is much more stochastic than suggested here. Hence, the true dates of seeding are likely to be later than these results suggest. Early December is possible, but the evidence here is not strong. R could have been higher and the variance of the offspring distribution could have generated seeding much later than is described here. It was not clear to me if the authors had looked at higher variance offspring distributions and if so, did they increase uncertainty in the timing of take-offs in the different locations.

Also - there were no comparative results for Asia. I would have thought that exactly the same model assumptions would have led to very early take-offs in Australia and New Zealand. How did

the border closures work there, with their connectivity to China, if the infection was so well seeded already in the US. It is possible that the high volume of travel to the US could have accounted for this, but I don't think it's the case. Definitely in Taiwan and Singapore? It does not seem reasonable to use a global model and only look at the timing of predicted times of take-off in one part of the world and not others. I suspect that the same model assumptions produce unrealistic take-off times in parts of the world not reported here.

The results here are of very high technical quality and the potential strengths and weaknesses of this approach versus the genomic approaches are also important. However, the claims here about early spread are important and on balance I do not think that the evidence is sufficiently robust to support them.

Detailed comments

Line 30: Over what period of time do these numbers refer to?

70: Please confirm that the source code is available and runs for this paper are easily reproducible.

Figure 1: These estimated averages are very smooth and must be conditional on the offspring distribution to some degree. Understanding that these are averages, conditional on the observed

124: The genomic epidemiology is really important for the primary hypothesis here and needs to be mentioned much earlier in the piece.

131: How is sustained transmission defined?

160: Clinical records and serology should allow us to pick-up cryptic infections.

168: Please define the contribution from one country to another more clearly. Is it the proportion of exported cases *_prior_* to passing a threshold where transmission is assumed to be self sustaining? Or is it a probabilistic assessment of which country may have led to sustained transmission, calculated for each run individually, that takes into account the timing of the importations relative to the start of sustained transmission?

172: The estimated rates of importation to different US states are probably overly precise given the underlying uncertainties of the model.

187: I do not agree with the assessment that this model can reasonably be expected to be more accurate than the coalescent model for estimating ancestral relationships. (see Above)

Figure 3: These are not model projections, they are model fits. Data are only present in the top plots. The lower plots are only projections for infection attack rates and they seem very high. There are many serological surveys for these countries from time periods not longer after the times presented here and these numbers look to be far too large. Does Ref 28 include any serology results or is it also a model fit to death data?

Figure 5: The model is pretty much constrained such that the rank ordering of epidemic size will be close to correct because of the correlation between the epidemic taking off and the time-to-lockdown. The correlation between values of attack rates (panel D) is shown on a log-log plot with a letter-box aspect ratio. Why are all geographies not shown on this plot - what was the method for finding serological data? How were differences in the timing of serology accounted for when matching with the model?

Author Rebuttals to Initial Comments:

Referees' comments:

Referee #1

We thank the referee for their careful reading of the manuscript and the many constructive comments.

Major comment:

1) Explaining 1 pandemic realization versus drawing general conclusions about the international spread of pandemics: I see three major limitations of the current manuscript

a. The key finding highlighted by the paper is that there was cryptic transmission in Europe and the US in the early stage of the pandemic. This is interesting but has already been documented in numerous studies; so this is not really a novel finding.

As the reviewer can see from the comments of other reviewers, the timing and extent of the cryptic phase does not appear as well established, and is even considered a “major claim” by one of the other reviewers. Indeed, while there are papers that have pointed out the early introduction and transmission of SARS-CoV-2 in specific locations, there is much less evidence in the rest of the world. We have modified the Discussion to clarify this concept:

- Line 192: *The model presented here captures the spatial and temporal heterogeneity of the early stage of the pandemic, going beyond the single country-level reconstruction. It provides a mechanistic understanding of the underlying dynamics of the pandemic's interconnected evolution. Furthermore, rather than showing specific evidence for early infection in a few locations, our study aims at providing a statistical characterization and quantification of the initial transmission pathways at a global scale.*

b. The reconstruction of the trajectory of the pandemic as it happened is another finding highlighted in multiple figures of the paper. However, this is done with a model that is not calibrated to the whole data available but only to information about which countries were the first impacted.....it is hard to justify performing an analysis of all possible paths rather than the single observed one. I would say this is a major limitation of the paper in its current form.

Actually, we believe this is one of the strengths of the paper. The value of the modeling approach presented is indeed a statistical evaluation of the spatio-temporal pattern and does not incorporate information that is generally available after the fact (retrospectively, or after the detection of cases and isolation of virus samples). In our modeling exercise we only use the first set of imported cases from China (up to January 21, as detailed in the SI), human mobility data, and the real-world implementation of travel restrictions policies that are known in real-time. The presented results show that the model is able to generate a statistical description of the early spreading dynamics that is then confirmed by out-of-sample data. In other words, the mechanistic modeling approach has informative predictive power that can be harnessed to anticipate the spatio-temporal spread of a disease. This is what makes the approach useful as a real-time risk assessment and situational awareness tool. It does not compete with, but rather complements, the information provided by phylogenetic analyses and other surveillance methods. We have revised the Introduction and Discussion to clarify the complementary aspects of these different approaches and their aims.

c. Insights gained about control strategies for future pandemics remain relatively limited. Authors write: “reactive response strategies, such as issuing travel restrictions targeting countries only after local transmission is confirmed are highly inefficient.” However they have not used their model to test and compare different control strategy. So, while the current work is interesting, I do not think it makes best use of the metapopulation model that is being presented.....

We agree with the referee that providing a detailed analysis and comparison of different control strategies would be very valuable and that the structure of the model we adopt is particularly

well suited for this task. In the revised version of the manuscript, we have added a new analysis showing a counterfactual scenario based on a broader testing policy (see Section 8 of the SI). This new analysis shows the effects of testing schemes that can detect and isolate 50% of infections without limiting testing to specific travel histories. In this case, the epidemic timeline (measured as the probability of generating at least 10 cumulative infections per million inhabitants in each country) is considerably delayed (Figure S17, enclosed also below) with respect to the baseline scenario which assumes a detection rate as happened. In the figure the white star marks the date where the probability is at least 90%. While the overall detection of infections in each country is relatively small, the benefit appears relevant. On the other hand, it must be stressed that a broad testing strategy requires testing capacity to ramp up very quickly. We have included a full section on the counterfactual scenario in the SI of the manuscript.

2) I felt that the last part of the paper estimating Infection Fatality Ratios was disconnected from the rest of the paper (in this part there is additional calibration to more country data and it considers a longer time period than the early stage of cryptic transmission). My suggestion would therefore be to refocus the paper on international spread and implications for future pandemics, remove this part and add simulation studies of interventions at international level.

The aim of the research is to characterize the conditions, events, and patterns that lead to the spread and establishment of the first COVID-19 pandemic wave in the US and Europe as well as to estimate its burden. As such, we believe that the estimation of the infection attack rates (AR) and infection fatality ratios (IFRs) are important contributions of the paper. On one hand such estimates allow us to quantify the burden of the first pandemic wave in multiple world regions, including those where little data are available. On the other hand, it allows us to further validate the results with independent observations from the literature. Furthermore, these results are important in showing that the onset of local transmission and the timing mitigation policies are entangled together in determining the magnitude of the first wave in each country.

Minor comments:

Figure 1B: since deaths is a better proxy of SARS-CoV-2 circulation than cases, can authors compare observed and predicted number of deaths by location.

The referee is right that deaths, as we also acknowledged in the paper, are a better proxy of SARS-CoV-2 circulation. However, Figure 1 is focused on the early stage of the epidemic, when, unfortunately, deaths were not detected as well because SARS-CoV-2 circulation was still relatively low (as discussed in the paper). The reported numbers also serve as a point to highlight how the picture obtained from surveillance (Figure 1 A) was extremely far from capturing the spreading of the virus. We provide here the analysis presented in Fig. 1B but correlate the cumulative estimated infections with the total reported deaths by March 29, 2020. By this date, 27 of the 30 European countries analyzed and 47 US states reported at least 1 death. We still find a strong correlation between the estimated and reported values. The Pearson correlation coefficient is calculated on the logged values and the sizes of the circles are proportional to the population size of that particular country or state.

Line 154: "The wide distribution of dates suggests that SARS-CoV-2 cryptic transmission may have begun as early as December 2019." From Figure 2, it does not look like there was high likelihood of cryptic transmission in December. What is the associated probability?

Our comment was referring to the fact that the distributions of local onset for the first states/countries contain non-zero values for December 2019. However, as the referee correctly noted, the corresponding values are small. In the revised version of the manuscript we clarified this point and provide information about the associated probabilities:

- *Line 109: However, it is worth noting that each distribution $p(t)$ has a support spanning several weeks. For instance in Italy the 5th and 95th percentiles for the onset of local transmission are on January 6 and January 30, 2020. These dates also suggest that it is not possible to rule out introductions and transmission events as early as December, 2019, although their probability is very small.*

Figure 2: Isn't that redundant with Figure 1C? If authors feel it's not, then I think a map of that graph would be more informative so we can understand spatial patterns (for example, colour the country/state as a function of the date of onset).

While certainly the data shown in the two panels are linked, they describe different points. In figure 1C we show the probability that at least 100 local infections were generated in each catchment area of the model in Europe and US by a particular day (Feb 21). As mentioned above, the figure substantiates how the surveillance data drastically underestimated the community spread of the virus in the initial phases of the pandemic. In Figure 2, we show the distribution of local onset times defined as the times when at least 10 local infections were generated in each country/state. We believe it is important to show the full distributions of the onset times rather than a single value (e.g., the median) as we would have to if used a geographical representation of the data. Showing the full distribution, particularly the left tail pointing at the earlier onset dates, naturally sparks the discussion about possible early starts of community outbreaks in December 2019/January 2020.

Figure 3: Authors need to clarify in the legend over which time period the importation source is characterized. Overall, I do not find that the Figure 3 in the main text is very informative for the discussion about the early stage of the pandemic as it seems quite obvious that if we consider an extended period up to the end of April, SARS-CoV-2 will be everywhere and so most introductions will be from neighbouring areas. I feel that the equivalent of Figure 3 that is in the Supplement and shows where introductions come from at the start of local transmission, is actually much more interesting. For Europe for example, we see a clear signal where first impacted countries were infected by China, then Asian countries and then European countries. If I was to select one of these two figures, my preference would clearly go to Figure S9 (arguably, the 2 would fit in the main manuscript – but I don't find current Fig 3 that covers a much wider time period very informative – as it moves away from the period of cryptic transmission that is the focus of the paper). In terms of presentation: i) Since there are only 5 possible origins/colours, I feel the current presentation is overly complex. Results would be

clearer if they were represented with a barplot, with data for each state/country being presented with a single colored bar (1 color showing the proportion of the associated origin), and with the bars ordered by the date of introduction. ii) for colour coding, I think it would be clearer if the same colours were used for the same locations in the 2 panels (currently, Europe and the US switch colours between figures which I find confusing).

We thank the reviewer for the careful analysis of the figure. We agree that the plots provided in the main text and those provided in the SI are offering different and equally relevant information on the pandemic seeding and expansion across the world. We considered moving both figures to the main text, but the resulting figure was very hard to read and rather confusing. As a result, we had to make the choice of which illustration to feature in the main manuscript. We provide summary statistics on all introductions to enable direct comparison with several papers reporting genomic sequencing analysis, so that genomic outputs can be thus compared with our results. However, as the reviewer also suggests, we are specifically referring to the plots reported in the SI in the main text. Concerning the figure composition, the number of states and countries is large enough (79 locations) that small bar plots have the same limits as the currently used circular illustrations. As suggested by the reviewer, we have revised the color code to maintain consistent geographical labels across figures.

Concerning the air transportation data: if an individual goes from Asia to Italy with a transit in France, will they be counted as an importation from France? This needs to be clarified as this would represent an important limitation to the analysis.

We have the privilege to work with Origin-Destination data, instead of the single connection traffic data; therefore we do not have this limitation in our analysis. We make this clear in the revised version of the manuscript.

- Line 256: *The airline transportation data encompass daily travel data in the origin-destination format from the Official Aviation Guide (OAG) database reflecting actual traffic changes that occurred during the pandemic.*

Page 7, line 318: “reactive response strategies, such as issuing travel restrictions targeting countries only after local transmission is confirmed are highly inefficient.” I do not think authors have demonstrated this point (that is, they haven’t presented an evaluation of this specific aspect). First, the start of local transmission is defined by authors with a relatively small number of local infections. It is not clear that for such low numbers, the risk of exportation to other countries is important; so, it’s perfectly possible that waiting a bit to implement travel restrictions is actually ok. Second, it may just be that travel restrictions are not very effective at limiting the risk of importation.

Thanks for this remark that allows us to explain in more detail our take on this. We believe that the paper shows clearly that targeting only countries where the epidemic is evidently spreading is putting the response strategy always one step behind the epidemic. In Fig. 3 and Fig. S9 we show that most of the introductions come from countries that were not targeted by travel

restrictions, while the contributions from China are indeed modest because of the travel restrictions. Concerning the effectiveness of travel restrictions, we have investigated this issue with respect to the introductions from China in a previous paper (Chinazzi, M. et al, The effect of travel restrictions on the spread of the 2019 novel coronavirus (COVID-19) outbreak. *Science*, 368(6489), pp.395-400.).

Methods, line 357: "To estimate the number of deaths we use as prior the age-stratified infection fatality ratios from Ref 48." The use of a prior would make sense if authors were trying to estimate the infection fatality ratio knowing the number of deaths. Here, they write that the aim is to estimate the number of deaths. So, are the estimates from ref 48 a prior? Or do they assume IFRs equal to those in ref 48? This needs to be clarified.

We thank the reviewer for this remark that allows us to clarify this point. In the second part of the manuscript, we use the reported death data to estimate the IFR and the infection AR of each country. We used Ref. 48 as a prior and then calibrated our model results using the reported deaths in each country to estimate the posteriors of the IFR and infection AR. We have clarified this point in the revised text.

Does Figure 6 in the supplement show the correlation observed-expected number of deaths with the ABC step considering deaths data in individual countries (as the one used in Figure 4)? Or does it show the unconstrained simulations?

The correlation is between the weekly number of reported deaths with the results of the additional ABC step. This is to provide the goodness of fit attained by the ABC because in our approach the rejection criterion is on the wMAPE for the whole epidemic profile and NOT on the individual, weekly values. We added an explicit statement about the calibration method used to the figure caption in the SI.

Figure 4-6 in the supplement: "using the calibration reported in the main text". It's not clear which calibration authors are talking about since results for 2 types of calibrations are used in the main text of the article.

Thank you for pointing this out. In the new version we have clarified that we use the wMAPE state/country level calibration method with a 25% tolerance threshold. We added an explicit statement about the calibration method used to the figure captions in the SI.

Please label differently Figures in the Supplement (e.g. S1, S2...) and in the main text to avoid confusion.

We are grateful to the reviewer for pointing this out. We have relabeled the figures in the supplement accordingly.

In section 3.2 in the supplement, authors present an “Alternative distance measure for model calibration” where the ABC is also calibrated on deaths data. Is this similar to the method they used to estimate IFRs?

The method is the same, but we use a different acceptance criterion to show the robustness of the ABC approach. We clarify this in the revised SI.

Referee #2 (Remarks to the Author):

This is an important study of the early transmission dynamics of COVID-19 and will be a seminal contribution to the literature when published. I think it is important to distinguish what /could have happened/ (from simulations) with /what actually happened/, which I think authors have done a good job of this. Since you have a global model, why focus on US and Europe but not considering other locations in Asia, or Africa and South America? Iran was also very heavily affected in the early days, if I remember correctly.

We are truly glad that the reviewer considers our study a “seminal” contribution. We agree that given the global nature of the model, it is quite natural to think of what the model could tell about the rest of the world. We decided to focus on the US and Europe as countries where similar mitigation approaches were initially considered and where extensive information about the adopted response measures is available as well as studies of phylogenetic analyses that we use for validation. The discussion of the US and European experiences is already extensive and extending the same level of discussion to the whole world would not fit into a single paper. At the same time, we understand the importance of showing results for at least a subset of countries in other continents as a validation of the approach and its consistency beyond Europe and the US. This point is echoed also by reviewer #3. For this reason, we have added an entire section in the SI (Section 7) that considers 24 additional countries: Argentina, Australia, Canada, Chile, Colombia, Egypt, Ethiopia, India, Indonesia, Japan, Kenya, Korea, Rep., Malaysia, Mexico, New Zealand, Nigeria, Philippines, Singapore, South Africa, Sudan, Taiwan, Tunisia, Turkey, and the United Arab Emirates. We provide estimates of the onset of local transmission and the burden of the first wave for these countries and highlight the differences between these additional countries and those in Europe and the US included in the main text. We briefly comment on these results in the main paper, but we defer a more detailed discussion of other regions of the world for future papers. Future analyses should also take into account issues related to NPIs, data quality, etc. in low and middle income countries. We hope that the new data and analyses add valuable material and validation to our modeling approach. Some of the figures added in the SI are reported in the reply to referee #3.

Minor comments:

Abstract - Infection attack rates of 0.78% to 15.2% and 0.19% to 13.2% - these are very wide ranges, could you name some of the locations with estimated higher or lower incidence to make it clearer the variability is in location-specific estimates?

Thanks for pointing this out. We now specify that these ranges refer to geographical heterogeneities and state explicitly in the abstract:

- *We find a heterogeneous, geographic distribution of cumulative infection attack rates by July 4, 2020, ranging from 0.78%- 15.2% across US states and 0.19%-13.2% in European countries.*

In line 136-7, you mention the extinction probability, does this take into account overdispersion (superspreading)?

The analytical estimate of the extinction probability calculation reported in the text does not account for overdispersion. We add this statement to the manuscript.

Referee #3 (Remarks to the Author):

We would like to thank the referee for the insightful comments and observations which have allowed us to improve the manuscript.

The authors present analysis from a well-known and validated global model of infectious disease transmission. They address the question of how quickly the virus spread initially and how much earlier many populations were seeded than was evident in the public case and death data. They conclude that there was likely widespread transmission in many populations much earlier than is currently assumed. This would be a major claim if published here.

We agree with the reviewer that the claim of cryptic transmission in many countries is one of the main results of our paper. In the scientific literature there are claims of local transmission (coming from analysis of blood banks, sewage, etc., see citations in the introduction of the paper and below) that cover a very broad range of dates, some dating back as early as Fall 2019. We believe that our paper provides a statistical estimate of the plausible time window for early establishment as well as epidemic pathways for a large number of countries. As we see from the contrasting opinions of this reviewer with reviewer #1 (who considers the cryptic stage as well established), this is a subject of considerable debate. As a result, we are reinforced in our belief that this paper represents a valuable contribution to our understanding of the early phase of the pandemic.

Some key results are compared with those from the coalescent model that is used by evolutionary virologists to address the same question. Where the models disagree, the authors point to potential biases in the coalescent work. Although neither approach is free from bias, the evolutionary models make use of much more data in combining dates and locations of whole genomes. The epi model here can be thought of as a forward simulation of the linked exponential processes assumed when fitting the phylogeographic models (or looking at the tip locations in non-geographic phylogenetic trees). Also, it could be argued that variable growth rates in the coalescent model are more flexible for this type of question.

It is worth stressing here that in no way do we intend to propose our approach as a competing or better approach than the coalescent model or any other phylogenetic analysis.

We believe that our approach is complementary and based on very different data inputs. We use a mechanistic approach that aims at simulating the evolution of the pandemic on the basis of early epidemiological data from the outbreak, thus providing statistical information on the likely evolution before genomic data can be collected in the field. Indeed, we use the phylogenetic analysis only as a validation and not as input to our model. Differently from phylogenetic analyses, our approach does not attempt an exact reconstruction of a single pandemic realization, but rather provides insight on the likely pandemic trajectory that can be used for situational awareness and to guide testing and response strategies. Furthermore, it may provide insights in countries or locations that do not have sequencing capabilities. This is not meant to compete with the coalescent model, but rather provide additional information, cross-validation, and an approach that can be used in the absence of genomic data. We have now emphasized this in the main text to avoid any confusion on this point. In particular, the text has been revised as follows:

- Line 51: *Our model complements phylogenetic analyses and other surveillance approaches. It maps the plausible pathways of the pandemic using information available in the early stages and provides a global picture of the cryptic phase as well as the ensuing first wave of the COVID-19 pandemic.*

The underlying process of global spread and establishment is much more stochastic than suggested here. Hence, the true dates of seeding are likely to be later than these results suggest. Early December is possible, but the evidence here is not strong. R could have been higher and the variance of the offspring distribution could have generated seeding much later than is described here. It was not clear to me if the authors had looked at higher variance offspring distributions and if so, did they increase uncertainty in the timing of take-offs in the different locations.

We agree with the reviewer on this point and indeed we believe that our figure 2 conveys this message. The support of the distribution for the onset of local transmission generally spans several weeks (starting in December 2019). While the onset time probability is very low in December 2019, we cannot rule out early local transmission events, the support for the distribution is generally over a span of a few weeks. This is why we believe that showing the full distribution is more appropriate than a central estimate. We have now provided a more detailed discussion of this point to highlight the stochasticity of the establishment of the pandemic in different locations.

- Line 108: *However, it is worth noting that each distribution, $p(t)$, has a support spanning several weeks. In Italy, the 5th and 95th percentiles of the $p(t)$ distribution are January 6 and January 30, 2020. These dates also suggest that it is not possible to rule out introductions and transmission events as early as December, 2019, although the probability is very small.*

Also - there were no comparative results for Asia. I would have thought that exactly the same model assumptions would have led to very early take-offs in Australia and New Zealand. How did the border closures work there, with their connectivity to China, if the infection was so well seeded already in the US. It is possible that the high volume of travel to the US could have accounted for this, but I don't think it's the case. Definitely in Taiwan and Singapore? It does not seem reasonable to use a global model and only look at the timing of predicted times of take-off in one part of the world and not others. I suspect that the same model assumptions produce unrealistic take-off times in parts of the world not reported here.

We thank the reviewer for raising this point. This comment and the suggestion of the reviewer #2 have prompted us to add a specific section in the SI, where we provide an analysis of the onset of local transmission and burden estimate for the first wave in 24 additional countries in several world regions including Latin America, Middle East, East Asia, Africa, and Oceania. This brings the total number of countries analyzed to 50, plus the additional 49 continental US states. In particular, we are able to compare projections against observations for all the countries suggested by the reviewer.

We want to highlight here that while the onset of local transmission in East Asia was very early, other countries in Africa experienced an onset of local transmission much later. However, we have to distinguish the onset of local transmission from the evolution of the first wave. Indeed, the very aggressive, early mitigation approaches of countries like Japan, South Korea etc. have given rise to epidemic trajectories very different from those observed in the US and Europe. In other words, it is important to keep in mind that the onset of local transmission and the following evolution of the epidemic are two different aspects of the epidemic affected in different ways by interventions and policies initiated in each country. For this reason, in the SI (Section 7) we report a section that discusses the results for 24 countries outside of Europe and the US. In this new analysis, we estimate that the onset of local transmission is broadly distributed across countries, and that the impact of the epidemic in the different countries is very heterogeneous. In the figure below (Figure S14) we show the posterior distribution of the estimated onset of local transmission dates (A). Furthermore, we report the correlation of our estimates of the onset dates with real data (i.e., the date of the first reported death and the date where there were 50 total cases detected).

Additionally, we measure the burden on each of the countries above through estimating the infection AR and IFR using a similar state/country level calibration used for the individual US states and European countries with the reported deaths. In the figures below we show the model fits of the weekly death estimates for all 24 countries analyzed and the posterior distributions of the infection AR on July 4, 2020 and the country level IFRs. We believe that the model provides a good picture of the evolution of the pandemic in the different world regions. However, a full discussion of the differences across world regions and their mitigation approaches cannot fit in the space in a single manuscript. We, therefore, keep the focus of the main text on Europe and the US, while we report results from other countries in the SI as additional validity of our approach. We intend to tackle specific regional analyses in future papers.

Detailed comments

Line 30: Over what period of time do these numbers refer to?

The estimates are done considering weekly deaths in the various states/countries. The range reported in the abstract refers to the range of infection AR observed in different geographical locations (states or countries). The time window for infection AR is up to July 4, 2020, and it is the same for all countries. We have clarified this sentence in the abstract and in the main text.

70: Please confirm that the source code is available and runs for this paper are easily reproducible.

We have always made our model publicly available, supported and documented at the following web pages: www.gleamviz.org; <https://www.gleamproject.org/>.

The constitutive equations and algorithms of the model have been detailed in several publications: Proc Natl Acad Sci USA 106, 21484-21489 (2009); Journal of Computational Science. 1: 132 (2010); BMC Infectious Diseases, 11 :37 (2011); Proc Natl Acad Sci USA 114(22) (2017).

The model has been used previously by a number of other research teams (among others Nature communications 9.1 (2018): 218; Epidemics 21, (2017) Pages 1-12; BMC Infectious Diseases (2016) 16:70; Science 342.6164 (2013): 1337-1342).

The model has also been independently used by other teams in the context of the COVID-19 analysis: PeerJ 8:e9548 <https://doi.org/10.7717/peerj.9548>; Int. J. Environ. Res. Public Health 2020, 17(18), 6530; [The Effect of Strict State Measures on the Epidemiologic Curve of COVID-19 Infection in the Context of a Developing Country: A Simulation from Jordan; https://www.medrxiv.org/content/medrxiv/early/2020/06/05/2020.06.03.20121517.full.pdf](https://www.medrxiv.org/content/medrxiv/early/2020/06/05/2020.06.03.20121517.full.pdf)). We will provide at the time of publication a github repository with all the data used to generate each figure. We hope this might help with independent analysis and comparisons.

The model uses proprietary mobility data (OAG) that are commercially available and cannot be shared publicly, although they can be acquired through the data provider. All the other data used are publicly available.

Figure 1: These estimated averages are very smooth and must be conditional on the offspring distribution to some degree. Understanding that these are averages, conditional on the observed

We are not able to fully understand what the reviewer is referring to in Fig. 1. All the results are averaged over multiple runs. The model simulates individual level transmission and infections in each metapopulation area. Therefore, the distributions are the outcome of the average at the reported geographical resolutions of the many mechanistic transmission processes.

124: The genomic epidemiology is really important for the primary hypothesis here and needs to be mentioned much earlier in the piece.

We thank the reviewer for this comment. In the revised version we mention genomic epidemiology in the introduction of the paper and point to specific differences from our approach.

131: How is sustained transmission defined?

As described in lines 91-93 “we define the onset of local transmission for a country or state as the earliest date when at least 10 new infections are generated each day”.

160: Clinical records and serology should allow us to pick-up cryptic infections.

We agree with the reviewer. In the final sentence of that paragraph we aim to highlight how the study of the cryptic transmission phase and the characterization of the onset of national outbreaks has been (and will remain) a matter of study and persisting debate. Literature is conflicting on the starting dates of local transmission and several papers have been published using a variety of different methods. We have added these citations to our manuscript:

1. Keri N Althoff et al., Antibodies to SARS-CoV-2 in All of Us Research Program Participants, January 2-March 18, 2020, *Clinical Infectious Diseases*, 2021;, ciab519, <https://doi.org/10.1093/cid/ciab519>
2. Bedford, T. *et al.* Cryptic transmission of SARS-CoV-2 in Washington state. *Science* **370**, 571–575 (2020).
3. CDCMMWR. Evidence for Limited Early Spread of COVID-19 Within the United States, January–February 2020. *MMWR Morb Mortal Wkly Rep* **69**, (2020).
4. La Rosa, G. *et al.* SARS-CoV-2 has been circulating in northern Italy since December 2019: Evidence from environmental monitoring. *Sci Total Environ* **750**, 141711 (2021).
5. Deslandes, A. *et al.* SARS-CoV-2 was already spreading in France in late December 2019. *International Journal of Antimicrobial Agents* **55**, 106006 (2020).
6. Basavaraju, S. V. *et al.* Serologic Testing of US Blood Donations to Identify Severe Acute Respiratory Syndrome Coronavirus 2 (SARS-CoV-2)–Reactive Antibodies: December 2019–January 2020. *Clinical Infectious Diseases* (2020) doi:[10.1093/cid/ciaa1785](https://doi.org/10.1093/cid/ciaa1785).

Our results contribute to such discussion, providing a bound to the plausible temporal window for introduction and onset of local transmission in many countries. Noticeably, our mechanistic modeling approach does not use genomic, clinical, or serologic data, thus providing additional, independent information addressing the early seeding of the pandemic.

168: Please define the contribution from one country to another more clearly. Is it the proportion of exported cases _prior_ to passing a threshold where transmission is assumed to be self sustaining? Or is it a probabilistic assessment of which country may have led to sustained transmission, calculated for each run individually, that takes into account the timing of the importations relative to the start of sustained transmission?

For each threshold considered (either April 30, 2020 in the main text or the onset of the local outbreak in the SM), the introductions are computed considering, for each run, the number of seeding events happening before or at the threshold. We do not add any further conditions/constraints on the sources. Indeed, while seeding events from sources that did not yet pass the conditions set for sustained local spreading are possible in principle, they are extremely unlikely considering the small numbers involved and the stochastic nature of the model.

172: *The estimated rates of importation to different US states are probably overly precise given the underlying uncertainties of the model.*

As we showed in the tables detailed in the SI (Tab. S8), the large number of runs considered introduce indeed ample margins of uncertainty defined around the averages shown in the figures.

187: *I do not agree with the assessment that this model can reasonably be expected to be more accurate than the coalescent model for estimating ancestral relationships. (see Above)*

As we detailed above, our intention was not to claim any type of methodological superiority. In fact, we strongly believe in the importance of considering both approaches, which nicely complement and validate each other. We did not want to suggest an increased accuracy for this model. Actually, even the way we discuss differences about the approaches are based on the different statistical point of views. We have changed the language in the revised manuscript.

Figure 3: These are not model projections, they are model fits. Data are only present in the top plots. The lower plots are only projections for infection attack rates and they seem very high. There are many serological surveys for these countries from time periods not longer after the times presented here and these numbers look to be far too large. Does Ref 28 include any serology results or is it also a model fit to death data?

We believe here the reviewer refers to figure 4 rather than figure 3, as figure 4 is where we discuss the IFR. We agree with the reviewer that these are posterior estimates coming from our ABC approach and not projections. We changed the terminology accordingly.

Concerning serology, we have used many of the surveys carried out during the first wave. These are reported in Figure 5D and Table S8 as also noted by the reviewer in the following. We also find that most empirical seroprevalence estimates fall within the 90%CI range of our model estimates. Here are a few examples:

- Denmark: 2.1% serological prevalence [1]; 1.05% [90CI 0.71-2.70] model estimated AR
- France: 4.9% serological prevalence [2]; 4.60% [3.33-9.95] model estimated AR
- San Francisco, CA: 1% serological prevalence [3]; 1.97% [0.51-6.54] model estimated AR

All model results are estimated as of the last day of the study period. Reference 28 in the main text estimates the serological prevalence for a number of countries and compares their results to different studies.

1. Erikstrup, C. *et al.* Estimation of SARS-CoV-2 Infection Fatality Rate by Real-time Antibody Screening of Blood Donors. *Clinical Infectious Diseases* **72**, 249–253 (2021).
2. O’Driscoll, M. *et al.* Age-specific mortality and immunity patterns of Sars-CoV-2. *Nature* **590**, 140–145. (2021).
3. Havers, F. P. *et al.* Seroprevalence of Antibodies to SARS-CoV-2 in 10 Sites in the United States, March 23-May 12, 2020. *JAMA Intern Med* (2020) doi:[10.1001/jamainternmed.2020.4130](https://doi.org/10.1001/jamainternmed.2020.4130).

Figure 5: The model is pretty much constrained such that the rank ordering of epidemic size will be close to correct because of the correlation between the epidemic taking off and the time-to-lockdown. The correlation between values of attack rates (panel D) is shown on a log-log plot with a letter-box aspect ratio. Why are all geographies not shown on this plot - what was the method for finding serological data? How were differences in the timing of serology accounted for when matching with the model?

The calibration steps constrain and select among possible epidemic paths. The various NPIs put in place globally certainly affect the spreading patterns. We opted for scatter plots rather than geographical representation to highlight the correlations across the board. The serological data are compared with the model estimates at the same collection time point as the serology survey. All the dates and serological references have been reported in the SI. Unfortunately, empirical estimates of seroprevalence at the population level are not available for all geographies considered in the paper. We can correlate only with the values reported in peer reviewed publications. We have now provided a clearer discussion of these points in the text and figure caption.

Reviewer Reports on the First Revision:

Referee #1 (Remarks to the Author):

I think that clarifications made to the manuscript have improved it. While the observation that transmission was cryptic has already been made for many locations, I appreciate that the systematic evaluation performed here is an important contribution to the literature.

Referee #2 (Remarks to the Author):

No further comments

Referee #3 (Remarks to the Author):

I thank the authors for their careful consideration of many of the points raised by the reviews. The manuscript is improved and reflects a substantial volume of very high quality theoretical epidemiology. The clarity of the text and figures all the way through to the end of the SI is of the highest standard.

However, I maintain some substantial concerns over the degree to which the results support the central claims.

Firstly, my comments about the variance assumed in the offspring distribution were not addressed. I have searched the main text and SI for any text related to the offspring distribution and I cannot find any: "offspring", "dispersion", "k-value". The model is reproducing a branching process that can be described by an offspring distribution. That distribution has a variance that is sometimes parameterized with a k-value or using the term dispersion. The variance of the offspring distribution will greatly affect these results, especially because they are conditioning on a single realization of a stochastic process. So even the modal posterior density of the time of takeoff in any country or state will likely be highly sensitive to the variance of the offspring distribution. If the variance is higher, seeding can be later. I suspect that the offspring distribution assumed here has a relatively low variance. The gleam model has been used to look at exactly this point in a prior PNAS paper on flu.

The importance of the offspring distribution was first highlighted in Nature in a very highly cited article about SARS-CoV-1 <https://pubmed.ncbi.nlm.nih.gov/16292310/>

I believe that the low variance in the offspring distribution drives the unrealistically early estimates of take off. Take off is defined as 10 infections _per day_, which means many more cumulative infections prior to that plus failed chains of transmission. From 10 infection per day, the probability of fade out for SARS-CoV-2 without behaviour change is very low. The _modal_ values for Australia, Singapore, Taiwan and elsewhere are not realistic. No interventions were in place in mid January in these places, nor was there widespread concern about the presence of the virus. This paper is confidently reporting modal posterior densities for times of takeoff as a central result. When there is such clear inconsistency in the trend of the country modal values, the authors should not point to wide credibility intervals and 2 or 3 serological studies for which no explicit attempt was made to account for assay sensitivity, as justification for maintaining their central claims.

Finally, there is a line added to the methods in the track changes version about the model seeding being consistent with circulation in China in October. I searched for any mention of this in the rebuttal document and didn't find any. It may be that the virus circulated in October in China. Or it may be that it really was only the very beginning of December that it made it to humans and the TMRCA from the genomic studies are actually estimating events in reservoir hosts. Or it may be somewhere in between. However, given the observations above this work provides zero evidence one way or the other. I do not know if this is meant to be interpreted as a claim in the methods section or as an opinion, but given the sensitivity around this issue, I am surprised it was added without comment.

Author Rebuttals to First Revision:

Detailed reply to the reviewers

Reviewer #1 & #2

We thank the reviewers for their careful reading of the manuscript and for considering it an important contribution to the literature.

Reviewer #3

We are truly honored by the opening statement of the reviewer that considers the *“manuscript is improved and reflects a substantial volume of very high quality theoretical epidemiology. The clarity of the text and figures all the way through to the end of the SI is of the highest standard”*

Seldom have we encountered such kind words during the review process.

At the same time the reviewer expresses some concerns related to the central claims of the paper. Let us first address what we think is the main concern of the reviewer, summarized in the reviewer’s consideration that our model provides “unrealistically early estimates of take off”.

We believe that all other concerns and remarks stem from this judgement. Our paper points to central estimates for the onset of local transmission generally spanning from late January to late February. The onset of local transmission is defined as ten new infections per day in very large populations of hundreds of thousands or millions of individuals (US states/European countries). Thus, we are talking about early clusters with ongoing transmission. For instance, in the US, the UK, and Italy this timeline is supported by several papers (based on sequencing data and phylogenetic analysis, or clinical cases) that are listed in the discussion below. Furthermore, the CDC recently updated the official list of *Provisional Death Counts for Coronavirus Disease 2019*, adding a number of fatalities recorded in early January (the weeks of January 11th, 18th, and 25th more precisely) across different US states (<https://www.cdc.gov/nchs/nvss/vsrr/covid19/index.htm>). In other words, most of the literature provides evidence that is in agreement with our findings. Perhaps some of the reviewer’s concern originated from our use of the phrase “widespread transmission” in a few sentences of the paper. Since we were referring to 10 new transmission events per day in an entire country or state, this could lead to the general misunderstanding that our central estimates were referring to a full blown epidemic. This is not the case, and we have now made it clear throughout the paper that we refer to the onset of local transmission.

In the following we want to address in more detail some of the remarks related to the above concern of the reviewer.

- a) *“When there is such clear inconsistency in the trend of the country modal values, the authors should not point to wide credibility intervals and 2 or 3 serological studies for which no explicit attempt was made to account for assay sensitivity, as justification for maintaining their central claims.”*

We respectfully disagree with this statement from the reviewer. We support our results by citing throughout the manuscript a number of papers (more narrow in the geographic scope- Refs [30, 31, 42]) based on genetic sequences and phylogenetic analysis that are in line with our estimates. Furthermore, we are aware that some of the serological studies stirred quite a controversy. Indeed, in most of the cases our results tend to rule out the results of those studies as “too early take off”.

Also, there are several other recent and less recent papers that are not among the citations (the paper is not intended as a review of phylogenetic results) that support our findings.

Among those:

- I. The emergence of an early SARS-CoV-2 epidemic in the United States (<https://www.medrxiv.org/content/10.1101/2021.02.05.21251235v1>)
Location: Louisiana
Main Finding: The posterior median TMRCA of the Louisiana clade was February 13th , suggesting that low levels of local SARS-CoV-2 transmission within Louisiana were likely already ongoing prior to Mardi Gras 2020 (Feb. 25). *Our central estimate for the onset of local transmission (10 infections /day) is 2/23*
- II. Molecular evidence of SARS-CoV-2 in New York before the first pandemic wave (<https://www.nature.com/articles/s41467-021-23688-7.pdf>)
Method: respiratory pathogen-negative nasopharyngeal specimens from 3,040 patients across the Mount Sinai Health System in New York + phylogenetic analysis
Location: New York City
Main Finding: Evidence of sporadic SARS-CoV-2 infections a full month before both the first officially documented case and emergence of New York as a COVID-19 epicenter in March 2020.
- III. Genomic epidemiology of SARS-CoV-2 reveals multiple lineages and early spread of SARS-CoV-2 infections in Lombardy, Italy
<https://www.nature.com/articles/s41467-020-20688-x>
Location: Lombardy, Italy
Main Finding: Transmission during the second half of January, in line with evidence of initial outbreaks in other European countries in the second half of January.
- IV. Phylogeography of SARS-CoV-2 pandemic in Spain (<http://www.zoores.ac.cn/en/article/doi/10.24272/j.issn.2095-8137.2020.217>)
Location: Spain
Main finding: introduction of SARS-CoV-2 in the country could have taken place in the form of a B3 genome of unknown geographic origin, which mutated in Spain into B3a around 11 February (95% CI: 30 January to 20 February).
- V. Genomic characterization and phylogenetic analysis of SARS-COV-2 in Italy (<https://onlinelibrary.wiley.com/doi/10.1002/jmv.25794>)
Location: Northern Italy
Main Finding: data suggest that SARS-CoV-2 virus entered northern Italy between the second half of January and early February 2020

b) *“The _modal_ values for Australia, Singapore, Taiwan and elsewhere are not realistic. No interventions were in place in mid January in these places, nor was there widespread concern about the presence of the virus.”*

It is first important to consider the school calendar of different countries. While school closures during January/February were mostly due to summer breaks/holidays, they limited contacts between children in the early phase of the pandemic in specific regions. Here are a few details concerning school closures if we consider the list of countries provided by the reviewer:

- Australia: Schools are closed from December 14, 2019 (approx, varies by region) to the end of January/beginning of February 2020 for the summer break <https://info.australia.gov.au/about-australia/special-dates-and-events/school-term-dates>
- New Zealand: Schools are closed from December 20, 2019 to January 27/ February 7, 2020 (approx, varies by region) for summer break (<https://www.education.govt.nz/school/school-terms-and-holiday-dates/school-terms-and-holidays-archive/#twentytwenty>)
- Taiwan: Schools are closed from January 23 to January 29 of 2020 according to the calendar (Chinese New Year, <https://www.timeanddate.com/calendar/?year=2020&country=71&wno=1>). The holiday break was then extended until February 25 because of SARS-CoV-2 (<https://focustaiwan.tw/society/202002020016>).

Additionally, though mid-January was very early in the course of the pandemic, there was awareness on the ground. For instance, we can compare the stringency index as defined by the The Oxford Coronavirus Government Response Tracker (OxCGRT) project of those countries with the index of the US. This indicates that there were interventions or mitigation measures in place as early as mid-January.

This index clearly shows the different levels of response.

In Singapore clusters were detected starting in early February (in our model the onset of local transmission is in the second half of January), but strict contact tracing was immediately implemented, along with mandatory mask wearing, etc. (government responses are available on multiple government websites and news outlets).

A more detailed approach to the early response of Taiwan and New Zealand are available here:
[https://www.thelancet.com/journals/lanwpc/article/PIIS2666-6065\(20\)30044-4/fulltext](https://www.thelancet.com/journals/lanwpc/article/PIIS2666-6065(20)30044-4/fulltext)

A brief summary of the enacted interventions is:

- Passenger screening in Taiwan: “This screening was eventually extended to all passengers entering Taiwan from high risk areas/countries in late January, and eventually extended to **all passengers regardless of their location of origin** in early February”
- A distinct feature of the Taiwan response was widespread use of face masks to reduce transmission from infected people (regardless of symptoms) as well as providing protection for wearers (mass masking).
- Taiwan: in February 2020, the start of the new semester for schools and high schools was delayed for two weeks.
- New Zealand: introduced border restriction to non-citizens/permanent residents from February onwards.

Many of those mitigation measures, including school closures, indicate that while local transmission started at an early time, the net reproduction number, the doubling time, the number of multiple introductions, and the level of mitigation efforts were extremely different thus leading to much more controllable epidemic outbreaks.

This is why we believe that, as we state in our paper, each country deserves a specific discussion that takes into account the multiple epidemic drivers and the timeline of the intervention policies. It is not appropriate to oversimplify descriptions of the interventions and alertness of these countries.

c) “The model is reproducing a branching process that can be described by an offspring distribution. That distribution has a variance that is sometimes parameterized with a k-value or using the term dispersion.”...“My comments about the variance assumed in the offspring distribution were not addressed. I have searched the main text and SI for any text related to the offspring distribution....”

In the first reply to the reviewers’ comments, we underlined that we are not working with a simple branching process. Our model is discrete, stochastic, individual based, and includes the following heterogeneities:

- Age structure of each subpopulation
- Heterogeneity of contacts by settings and location
- Heterogeneity of contacts by age and location
- Heterogeneity of travel patterns by age and location
- Different reproductive numbers by location
- Different mobility (local and global) by location

All the details are reported in the supplementary information. The above heterogeneities lead to distributions of individual reproductive numbers that are overdispersed relative to the Poisson distribution that is generated when infectious individuals, settings, and populations are homogeneously distributed (see for instance <https://royalsocietypublishing.org/doi/10.1098/rsif.2006.0185>). Given these individual

heterogeneities, the offspring distribution will depend on the specific details of each population, mobility constraints, NPIs, etc. We included a discussion of this point in the SI.

While it is possible to impose a unique dispersion factor on single introduction events modeled by a branching process and/or stylized models, it is not straightforward to disentangle this from the several heterogeneities of our model. In our case, we should account for age and settings (for instance there is evidence of lack of dispersion in schools), low versus high mobility areas, etc.

It is also clear that although dispersion has been measured as a relevant feature in several specific studies, it is much less clear the relevance at the global dispersion level where there are repeated introductions of hundreds of cases. Most results on super-spreading events are potentially altered by mitigation measures and observation biases, and it is thus not guaranteed that these measurements would also hold when few or no mitigation measures were in place and/or in case of wider community spread. Indeed, studies (including one by some of the authors of the paper) report dispersion factors that vary considerably, with large confidence intervals, and are dependent on the level of interventions (<https://www.science.org/doi/10.1126/science.abe2424>, [https://www.thelancet.com/journals/laninf/article/PIIS1473-3099\(20\)30287-5/fulltext](https://www.thelancet.com/journals/laninf/article/PIIS1473-3099(20)30287-5/fulltext)). Other limitations in these studies concern the lack of identification of asymptomatic spreading, underreporting of cases, biases on large events due to surveillance and NPIs, etc. For this reason, we avoided plugging in a specific overall dispersion factor over the already complex dynamic of our model without the possibility of differentiation across age brackets, settings, locations etc. We believe that such a parsimonious approach introduces less uncontrolled effects.

d) The reviewer is also concerned that “Finally, there is a line added to the methods in the track changes version about the model seeding being consistent with circulation in China in October.”

This sentence was originally in the Results section of the first submission draft; however, when restructuring the paper for the second submission, it was moved to the Methods section. The sentence is just meant to say that our model has a posterior distribution for the start of the outbreak in China that includes the possibility of transmission starting as early as October. This is common and well established in the literature and phylogenetic analysis suggests a non-zero probability to a start of the epidemic in mid/late October (see for instance <https://science.sciencemag.org/content/372/6540/412> and <https://www.sciencedirect.com/science/article/abs/pii/S1567134820301829?via%3Dihub>).

Reviewer Reports on the Second Revision:

Referee #1 (Remarks to the Author):

Referee 3 is making an important point: in a simulation model such as the one used by the authors, heterogeneity in the onset of local transmission in different countries will depend on the overdispersion in the offspring distribution, with high overdispersion corresponding to scenarios with super-spreading events. When transmission is affected by super-spreading events, we expect that the timing of emergence in different countries may be much more heterogeneous than in the absence of superspreading events since in the former scenario, there is an important 'chance factor' that means that 2 countries next to each other with similar characteristics could see radically different initial trajectories if one is affected by a super-spreading event but not the other. In contrast, we would not expect such differences in scenarios where overdispersion is limited. The model anticipates that heterogeneity in the onset of local transmission was limited, for example spanning across a 40 day period (19 Jan-1 Mar) in Europe. Referee 3 is concerned that this result could simply be explained if the overdispersion that was implicitly assumed in the model was too low compared to what is known about SARS-CoV-2. Shall overdispersion be larger than what is assumed in the model, it is possible that the timing of onset of local transmission would span a wider time period.

This is a valid and important point and I did not understand why the authors did not do more to address it in the manuscript where I could not find any mention of the referee's point nor any attempt to acknowledge, address or discuss the problem. It seems essential authors revise the manuscript to correctly account for this comment:

- 1) At the very least this deserves a solid paragraph in the discussion, discussing how overdispersion could impact the results.
- 2) In their response, authors indicate that due to the complex structure of their model, there is overdispersion in their offspring distribution. However, they do not provide any characterization of their offspring distribution. In such a stochastic simulation model, it should be relatively straightforward for authors to compute for a given country at a given time, the offspring distribution (i.e. probability that a case will generate 0, 1, 2,... cases) and from that characterize overdispersion. These results could be reported in the Supplement and inform the discussion paragraph. In particular, it will be important to compare overdispersion in the model with available estimates from the literature.
- 3) If overdispersion is lower in the model than what is suggested for SARS-CoV-2, one way to definitely address the concerns of the referee would be to do a sensitivity analyses where

overdispersion is increased in simulations. Would that be technically possible? This would help understand sensitivity of results to implicit assumptions about overdispersion.

As a general comment about the review process: when reading the authors' response to the referee, I find it very confusing that they extract a short sentence from the referee's comments as we do not know what has been left out. I recommend authors use the standard practice of copy-pasting the whole comment of the referee to improve and facilitate the review process.

Referee #3 (Remarks to the Author):

Its clear that the authors and I are not communicating effectively at this point. I'm sorry if I have not been clear in previous comments, and would re-iterate again that in terms of implementation and presentation this is a manuscript of the very highest standard.

However complex the simulation, it boils down to a branching process because the number of infectious individuals is so small compared to the number of susceptible people. I believe the variance of the implied branching process is unrealistically low and that this drives early inferred emergence times. There is a massive literature on the variance of the offspring distribution for SARS-2. Although the authors may feel it is out of scope, they could measure the distribution "empirically" from these simulations and show either a) it is consistent with observations or b) that their key results are not sensitive to the effective variance of the offspring distribution. The model structure cannot be taken as a more reliable way to estimate the offspring distribution than the many existing empirical studies. I don't know that my hypothesis is correct, but these claims should not be made without making sure this aspect of the model is consistent with observations.

The authors state in their rebuttal.

"The onset of local transmission is defined as ten new infections per day in very large populations of hundreds of thousands or millions of individuals (US states/European countries). Thus, we are talking about early clusters with ongoing transmission."

I don't really understand this. Because this is a model with a high R_0 (even with schools closed), 10 new infections per day is clearly in the deterministic regime and with 5 to 10 day doubling time, we would expect a) very low chance of fade out and b) very large total outbreaks prior to any widespread population change.

Simple TMRCA studies do not give reliable evidence of times of take-off because importation often comes repeatedly from the same remote epidemic. Hence, a few separate introductions from Wuhan to Lombardy will give an early TMRCA in Lombardy which is actually describing average diversity in Wuhan when stuttering (high variance offspring distribution) imports began. It does NOT provide strong evidence that there was sustained early transmission as implied by these results. See for example Figure 5 in <https://www.science.org/doi/full/10.1126/science.abf2946>.

Overall, I agree that there was substantial cryptic transmission of SARS-CoV-2, but the precise timing of that transmission around the world is an important question for each individual country. For the reasons stated here and previously, I do not believe that this model is giving an accurate description of that process for most/ many places. If this is published in Nature, it will set a strong precedence for every single country / state for which a value is given.

Author Rebuttals to Second Revision:

Detailed reply to the Referees' comments:

Referee #1

Referee 3 is making an important point: in a simulation model such as the one used by the authors, heterogeneity in the onset of local transmission in different countries will depend on the overdispersion in the offspring distribution, with high overdispersion corresponding to scenarios with super-spreading events. When transmission is affected by super-spreading events, we expect that the timing of emergence in different countries may be much more heterogeneous than in the absence of superspreading events since in the former scenario, there is an important 'chance factor' that means that 2 countries next to each other with similar characteristics could see radically different initial trajectories if one is affected by a super-spreading event but not the other. In contrast, we would not expect such differences in scenarios where overdispersion is limited. The model anticipates that heterogeneity in the onset of local transmission was limited, for example spanning across a 40 day period (19 Jan-1 Mar) in Europe. Referee 3 is concerned that this result could simply be explained if the overdispersion that was implicitly assumed in the model was too low compared to what is known about SARS-CoV-2. Shall overdispersion be larger than what is assumed in the model, it is possible that the timing of onset of local transmission would span a wider time period. This is a valid and important point and I did not understand why the authors did not do more to address it in the manuscript where I could not find any mention of the referee's point nor any attempt to acknowledge, address or discuss the problem.

We agree with the reviewer and apologize for not having addressed these questions appropriately in the previous rounds. While in our previous reply we focused on the epidemic timeline and a comparison with other published analyses, we did not realize that the reviewer wanted us to provide an empirical measurement of the existing overdispersion within our model. We hope that our additional analysis and changes to the manuscript are sufficient and responsive to all of the reviewer's concerns.

It seems essential authors revise the manuscript to correctly account for this comment:

1) At the very least this deserves a solid paragraph in the discussion, discussing how overdispersion could impact the results.

As suggested by the reviewer we have added an entire section in the supplementary information tackling the analysis of the overdispersion in our model and a sensitivity analysis increasing overdispersion on the transmission dynamic.

2) In their response, authors indicate that due to the complex structure of their model, there is overdispersion in their offspring distribution. However, they do not provide any characterization of their offspring distribution. In such a stochastic simulation model, it should be relatively straightforward for authors to compute for a given country at a given time, the offspring distribution (i.e. probability that a case will generate 0, 1, 2, ... cases) and from that characterize overdispersion. These results could be reported in the Supplement and inform the discussion paragraph. In particular, it will be important to compare overdispersion in the model with available estimates from the literature.

As we discussed in the previous round of reviews, our model has multiple sources of heterogeneity, and the amount of overdispersion thus depends on location, time, transmissibility,

day of the week, etc. Yet we are able to quantify the extent of overdispersion, selecting specific settings as a case study. We empirically measured the overdispersion in our model by looking at 1,000 introductions in the Wuhan area for one week in November 2019 ($R_0=2.5$). To compare our observed results with the literature, we identified two papers that measure overdispersion by analyzing primary infections and contact tracing data on a large scale. The first one is Sun et al. (Science, 15 Jan 2021 Vol 371, Issue 6526, 2021) which analyzes 1,178 SARS-CoV-2–infected individuals and their 15,648 close contacts, and reports for the transmission distribution an overdispersion parameter $k=0.3$ (95% CI: 0.23 to 0.39). The second study from Bi et al. (Lancet Inf. Dis, 20, 911 (2021)) analyzes 391 SARS-CoV-2 cases and 1,286 close contacts, reporting an overdispersion parameter $k=0.58$ (95% CI: 0.35 to 1.18).

We report in the new section of the supplementary information Figure S9 inserted below that shows the percentage of secondary infections produced by the corresponding percentage of primary infections in our model. On top of this, we plot the same quantity for an offspring distribution that follows a negative binomial with an overdispersion parameter $k=0.3$, $k=0.6$, $k=2$ and a Poisson distribution, all with means of 2.5. It is readily observable that the model distribution is far from a Poisson branching process and is more closely aligned with the negative binomial distribution. In the first figure below (A) we report the overdispersion measured in Wuhan, China. In the main text we only include the results for Wuhan, but for the rebuttal letter we included the overdispersion measurement for Laramie, Wyoming during the week of January 13, 2020. This city was chosen because it varies in both population size (30K in Laramie vs. 11M in Wuhan) and commuting behavior which could affect the measured overdispersion. We have also investigated many other regions and while the results may vary slightly for locations across the world, the 70-25 rule provides the overall characterization of the GLEAM model's overdispersion (for $R_0=2.5$).

3) If overdispersion is lower in the model than what is suggested for SARS-CoV-2, one way to definitely address the concerns of the referee would be to do a sensitivity analyses where overdispersion is increased in simulations. Would that be technically possible? This would help understand sensitivity of results to implicit assumptions about overdispersion.

To allow for overdispersion on top of the existing heterogeneities in the model, we also performed a sensitivity analysis with overdispersion on the offspring distribution with parameter $k=0.3$ and $k=0.6$. We then conducted a full rerun of the model to yield distributions for the time of the onset of local transmission. As we discussed this is a strong assumption, indeed also in Sun et al., Science, 371, Issue 6526, (2021) it is shown how there is a general variability according to settings; moreover, measurements vary considerably across papers. However, as suggested by the reviewer, it is important to show the sensitivity of the model to increasing overdispersion. We report the quantitative analysis of the results for the case with $k=0.3$ in the supplementary information. The estimated difference across European countries and US states for the median onset times, T , (at least 10 new, locally-generated infections per day) is about 3 days later in the new analysis. We also measure a very modest increase of the standard deviation of the distribution for each country of about 2.5 days. These differences are practically not observable on the “epiweek” scale (7 days) over which real data are generally analyzed. These small observed differences can be understood by considering that the time for the onset of local

transmission is determined by the global circulation of SARS-CoV-2 that results in multiple repeated introductions of infectious individuals in locations at an increasing frequency over time because of the global growth of the epidemic. While overdispersion has a large effect on single introductions, its effect is reduced when averaging over many transmission events of increasing frequency over time (Althouse et al PLOS Biology, Vol 18(11), 2021)

In Sec. 3.3 of the supplementary information, we also show the full distribution of the onset of local transmission (10 locally-generated infections per day), $p(t)$, for the scenario with $k=0.3$ for all European countries and US states analyzed in the paper. We believe that this strengthens considerably the results contained in the paper and fully addresses the concerns of the reviewers.

As a general comment about the review process: when reading the authors' response to the referee, I find it very confusing that they extract a short sentence from the referee's comments as we do not know what has been left out. I recommend authors use the standard practice of copy-pasting the whole comment of the referee to improve and facilitate the review process.

We apologize that the previous response format caused confusion and now report in full all of the reviewers' comments.

Referee #3

Its clear that the authors and I are not communicating effectively at this point. I'm sorry if I have not been clear in previous comments, and would re-iterate again that in terms of implementation and presentation this is a manuscript of the very highest standard.

We are very grateful to the reviewer for the kind and positive assessment of our work and also apologize for the miscommunication. While in our previous reply we focused on the epidemic timeline and a comparison with other published analyses, we did not realize that the reviewer wanted us to empirically measure the existing overdispersion within our model. We hope that our additional analysis and changes to the manuscript have sufficiently addressed your concerns.

However complex the simulation, it boils down to a branching process because the number of infectious individuals is so small compared to the number of susceptible people. I believe the variance of the implied branching process is unrealistically low and that this drives early inferred emergence times. There is a massive literature on the variance of the offspring distribution for SARS-2. Although the authors may feel it is out of scope, they could measure the distribution "empirically" from these simulations and show either a) it is consistent with observations or b) that their key results are not sensitive to the effective variance of the offspring distribution. The model structure cannot be taken as a more reliable way to estimate the offspring distribution than the many existing empirical studies. I don't know that my hypothesis is correct, but these claims should not be made without making sure this aspect of the model is consistent with observations.

To fully answer the remaining concerns of the reviewer, we have performed both suggested analyses, namely: a) measure empirically the overdispersion in our model; b) perform a sensitivity analysis showing the effect on the variance of the distribution by imposing, within the model, an increased overdispersion of the offspring distribution (in all locations and at all times). We report here the same reply and materials we wrote in the reply to reviewer #1 who was asking for the same tests. The reviewer and the editor will excuse us if we are redundant, but this is for the sake of the readability of the replies to each reviewer.

As we discussed in the previous round of reviews, our model has multiple sources of heterogeneity, and the amount of overdispersion thus depends on location, time, transmissibility, day of the week, etc. Yet we are able to quantify the extent of overdispersion, selecting specific settings as a case study. We empirically measured the overdispersion in our model by looking at 1,000 introductions in the Wuhan area for one week in November 2019 ($R_0=2.5$). To compare our observed results with the literature, we identified two papers that measure overdispersion by analyzing primary infections and contact tracing data on a large scale. The first one is Sun et al. (Science, 15 Jan 2021 Vol 371, Issue 6526, 2021) which analyzes 1,178 SARS-CoV-2-infected individuals and their 15,648 close contacts, and reports for the transmission distribution an overdispersion parameter $k=0.3$ (95% CI: 0.23 to 0.39). The second study from Bi et al. (Lancet Inf. Dis, 20, 911 (2021)) analyzes 391 SARS-CoV-2 cases and 1,286 close contacts, reporting an overdispersion parameter $k=0.58$ (95% CI: 0.35 to 1.18).

We report in the new section of the supplementary information Figure S9 inserted below that shows the percentage of secondary infections produced by the corresponding percentage of primary infections in our model. On top of this, we plot the same quantity for an offspring distribution that follows a negative binomial with an overdispersion parameter $k=0.3$, $k=0.6$, $k=2$ and a Poisson distribution, all with means of 2.5. It is readily observable that the model distribution is far from a Poisson branching process and is more closely aligned with the negative binomial distribution. In the first figure below (A) we report the overdispersion measured in Wuhan, China. In the main text we only include the results for Wuhan, but for the rebuttal letter we included the overdispersion measurement for Laramie, Wyoming during the week of January 13, 2020. This city was chosen because it varies in both population size (30K in Laramie vs. 11M in Wuhan) and commuting behavior which could affect the measured overdispersion. We have also investigated many other regions and while the results may vary slightly for locations across the world, the 70-25 rule provides the overall characterization of the GLEAM model's overdispersion (for $R_0=2.5$).

To allow for overdispersion on top of the existing heterogeneities in the model, we also performed a sensitivity analysis with overdispersion on the offspring distribution with parameter $k=0.3$ and $k=0.6$. We then conducted a full rerun of the model to yield distributions for the time of the onset of local transmission. As we discussed this is a strong assumption, indeed also in Sun et al., Science, 371, Issue 6526, (2021) it is shown how there is a general variability according to settings; moreover, measurements vary considerably across papers. However, as suggested by the reviewer, it is important to show the sensitivity of the model to increasing overdispersion. We report the quantitative analysis of the results for the case with $k=0.3$ in the supplementary

information. The estimated difference across European countries and US states for the median onset times, T , (at least 10 new, locally-generated infections per day) is about 3 days later in the new analysis. We also measure a very modest increase of the standard deviation of the distribution for each country of about 2.5 days. These differences are practically not observable on the “epiweek” scale (7 days) over which real data are generally analyzed. These small observed differences can be understood by considering that the time for the onset of local transmission is determined by the global circulation of SARS-CoV-2 that results in multiple repeated introductions of infectious individuals in locations at an increasing frequency over time because of the global growth of the epidemic. While overdispersion has a large effect on single introductions, its effect is reduced when averaging over many transmission events of increasing frequency over time (Althouse et al PLOS Biology, Vol 18(11), 2021)

In Sec. 3.3 of the supplementary information, we also show the full distribution of the onset of local transmission (10 locally-generated infections per day), $p(t)$, for the scenario with $k=0.3$ for all European countries and US states analyzed in the paper. We believe that this strengthens considerably the results contained in the paper and fully addresses the concerns of the reviewers.

Finally, let us stress that we have added to the supplementary information a full section addressing overdispersion and the sensitivity analysis. The section also reports the full onset distributions, $p(t)$, for the US states and European countries.

The authors state in their rebuttal.

"The onset of local transmission is defined as ten new infections per day in very large populations of hundreds of thousands or millions of individuals (US states/European countries). Thus, we are talking about early clusters with ongoing transmission."

I don't really understand this. Because this is a model with a high R_0 (even with schools closed), 10 new infections per day is clearly in the deterministic regime and with 5 to 10 day doubling time, we would expect a) very low chance of fade out and b) very large total outbreaks prior to any widespread population change.

Our point in the previous reply is that the onset of local transmission (10 locally-generated infections per day) corresponds to a very small number of infections. In some of the countries mentioned by the reviewer (Taiwan, New Zealand, etc.) this small number of infections was met by early mitigation measures, and thus it was easier to initially control the outbreaks contrary to what happened in Europe and the US where mitigation measures did not start before the end of February or early March.

Simple TMRCA studies do not give reliable evidence of times of take-off because importation often comes repeatedly from the same remote epidemic. Hence, a few separate introductions from Wuhan to Lombardy will give an early TMRCA in Lombardy which is actually describing average diversity in Wuhan when stuttering (high variance offspring distribution) imports began. It does NOT provide strong evidence that there was sustained early transmission as implied by these results. See for example Figure 5 in <https://www.science.org/doi/full/10.1126/science.abf2946>.

Many of the references we provided in our reply and the paper address specifically the time of onset of transmission. However, we agree with the reviewer, and this is one of the main reasons why we believe our paper is a relevant addition to those analyses as it is based on different data and approaches, thus providing a complementary point of view to existing analysis in the literature.

A) Empirical measurement of overdispersion in the model. We simulated for the geographical areas of Wuhan (China) and Laramie, Wyoming (US), with an $R_0=2.5$, 1,000 introductions and calculated the proportion of transmission attributable to a proportion of infections. The shaded area represents the uncertainty measured through a bootstrapping procedure that generates 100 sample distributions of 500 introductions over the full set of simulations. The GLEAM model shows that 25% of all primary infections are responsible for 70% of secondary infections within Wuhan. The other lines represent the proportion of transmission as a function of the proportion of infections in the case of exact Negative Binomial distributions with $k = 0.3, 0.6, 2.0$ (dotted lines) and the Poisson case ($k=\infty$) with mean 2.5 (solid line). **B), C)** Comparison of the distributions between the time of the onset of local transmission (10 locally-generated infections per day) of the GLEAM model (from the main text) and the model that uses, for all transmission events, a negative binomial offspring distribution with a dispersion parameter $k=0.3$. We calculate for each European country and US state considered in the study the distribution of the timing of local transmission $p(t)$, and report the difference of the median time T (B) and standard deviation (C). Specifically, we subtract the GLEAM model in the main text from the model with an overdispersion parameter $k=0.3$ for all locations (each data point is a European country or US state).

Timing of the onset of local transmission in the model where all individual transmission events follow a negative binomial with dispersion parameter $k=0.3$. Posterior distribution $p(t)$ of the week when each US state (A) or European country (B) first reached 10 locally generated SARS-CoV-2 transmission events per day.

Reviewer Reports on the Third Revision:

Reviewer #1 provided comments to the editor, declaring that if Reviewer #3 is satisfied they'd be happy for us to go ahead.

Referee #3 (Remarks to the Author):

I thank the authors for their rapid and comprehensive additional analyses to address previous comments. I am surprised that the variance in the offspring distribution doesn't make more of an impact on the time of establishment in the first affected remote locations, and I wonder a little about the weight in the right tail of the offspring distribution pdf that may be slightly masked by the presentation of the cdf. But I also accept the clarity of the additional evidence in comparing the delay difference between different distributions and that it is only a matter of days.